# COMPRESSION-AWARE TRAINING OF NEURAL NETWORKS USING FRANK-WOLFE

## ABSTRACT

Many existing Neural Network pruning approaches either rely on retraining to compensate for pruning-caused performance degradation or they induce strong biases to converge to a specific sparse solution throughout training. A third paradigm, 'compression-aware' training, obtains state-of-the-art dense models which are robust to a wide range of compression ratios using a single dense training run while also avoiding retraining. In that vein, we propose a constrained optimization framework centered around a versatile family of norm constraints and the *Stochastic Frank-Wolfe* (SFW) algorithm which together encourage convergence to well-performing solutions while inducing robustness towards convolutional filter pruning and low-rank matrix decomposition. Comparing our novel approaches to compression methods in these domains on benchmark image-classification architectures and datasets, we find that our proposed scheme is able to yield competitive results, often outperforming existing compression-aware approaches. In the case of low-rank matrix decomposition, our approach can require much less computational resources than nuclear-norm regularization based approaches by requiring only a fraction of the singular values in each iteration. As a special case, our proposed constraints can be extended to include the unstructured sparsity-inducing constraint proposed by Pokutta et al. (2020) and Miao et al. (2022), which we improve upon. Our findings also indicate that the robustness of SFW-trained models largely depends on the *gradient rescaling* of the learning rate and we establish a theoretical foundation for that practice.

## 1 INTRODUCTION

The astonishing success of Deep Neural Networks relies heavily on over-parameterized architectures (Zhang et al., 2016a) containing up to several billions of parameters. Consequently, modern networks require large amounts of storage and increasingly long, computationally intensive training and inference times, entailing tremendous financial and environmental costs (Strubell et al., 2019). To address this, a large body of work focuses on compressing networks, resulting in *sparse models* that require only a fraction of memory or floating-point operations while being as performant as their *dense* counterparts. Recent techniques include the *pruning* of individual parameters (LeCun et al., 1989; Hassibi & Stork, 1993; Han et al., 2015; Gale et al., 2019; Lin et al., 2020; Blalock et al., 2020) or group entities such as convolutional filters and entire neurons (Li et al., 2016; Alvarez & Salzmann, 2016; Liu et al., 2018; Yuan et al., 2021), the utilization of low-bit representations of networks *(quantization)* (Wang et al., 2018; Kim et al., 2020) as well as classical matrix- or tensor-decompositions (Zhang et al., 2016b; Tai et al., 2015; Xu et al., 2020; Liebenwein et al., 2021) in order to reduce the number of parameters.

While there is evidence of pruning being beneficial for the ability of a model to generalize (Blalock et al., 2020; Hoefler et al., 2021), a higher degree of sparsification will typically lead to a degradation in the predictive power of the network. To reduce this impact, two main paradigms have emerged. *Pruning after training*, most prominently exemplified by Iterative Magnitude Pruning (IMP) (Han et al., 2015), forms a class of algorithms characterized by a three-stage pipeline of regular (sparsity-agnostic) training followed by prune-retrain cycles that are either performed once (*One-Shot*) or iteratively. The need for retraining to recover pruning-induced losses is often considered to be an inherent disadvantage and computationally impractical (Liu et al., 2020; Ding et al., 2019; Wortsman et al., 2019; Lin et al., 2020). In that vein, *pruning during training* or *regularization* approaches avoid

retraining by inducing strong inductive biases to converge to a sparse model at the end of training (Zhu & Gupta, 2017; Carreira-Perpinán & Idelbayev, 2018; Kusupati et al., 2020; Liu et al., 2020). The ultimate pruning then results in a negligible performance degradation, rendering the retraining procedure superfluous. However, such procedures incorporate the goal sparsity into training, requiring to completely train a model per sparsity level, while IMP needs just one pretrained model to generate the entire accuracy-vs.-sparsity frontier, albeit at the price of retraining.

A third paradigm, which is the focus of this work, naturally emerges when no retraining is allowed, but training several times to generate the accuracy-vs.-sparsity tradeoff frontier is prohibitive. Ideally, the optimization procedure is "compression-aware" (Alvarez & Salzmann, 2017; Peste et al., 2022) or "pruning-aware" (Miao et al., 2022), allowing to train once and then being able to compress One-Shot to various degrees while keeping most of the performance without retraining (termed *pruning stability*). Compression-aware training procedures are expected to yield state-of-the-art dense models which are robust to pruning without its (regularization) hyperparameters being selected for a particular level of compression. While many such methods employ (potentially modified) SGD-variants to discrimnate between seemingly 'important'and 'unimportant'parameters, cf. GSM (Ding et al., 2019), LC (Carreira-Perpinán & Idelbayev, 2018), ABFP (Ding et al., 2018) or Polarization (Zhuang et al., 2020), actively encouraging the former to grow and penalizing the latter, an interesting line of research considers the usage of specific optimizers other than SGD. An optimization approach that is particularly suited is the first-order and projection-free *Stochastic Frank-Wolfe* (SFW) algorithm (Frank et al., 1956; Berrada et al., 2018; Pokutta et al., 2020; Tsiligkaridis & Roberts, 2020; Miao et al., 2022). While being valued throughout various domains of Machine Learning for its highly structured, sparsity-enhancing update directions (Lacoste-Julien et al., 2013; Zeng & Figueiredo, 2014; Carderera et al., 2021), the algorithm has only recently been considered for promoting sparsity in Neural Network architectures.

Addressing the issue of compression-aware training, we propose leveraging the SFW algorithm for a family of norm constraints actively encouraging robustness to convolutional filter pruning and low-rank matrix decomposition. Our approach, using the group-$k$-support norm and variants thereof (Argyriou et al., 2012; Rao et al., 2017; McDonald et al., 2016), is able to train state-of-the-art image classification architectures on large datasets to high accuracy, all the while biasing the network towards compression-robustness. Similarly motivated by the work of Pokutta et al. (2020) and, concurrent to our work, Miao et al. (2022) showed the effectiveness of $k$-sparse constraints, focusing solely on unstructured weight pruning. Our approach includes the unstructured pruning case as well, mitigating existing convergence and hyperparameter stability issues, while improving upon the previous approach. To the best of our knowledge, our work is the first to apply SFW for structured pruning tasks. In analyzing the techniques introduced by Pokutta et al. (2020), we find that the *gradient rescaling* of the learning rate is of utmost importance for obtaining high performing and pruning stable results. We lay the theoretical foundation for this practice by proving the convergence of SFW with gradient rescaling in the non-convex stochastic case, extending results of Reddi et al. (2016).

**Contributions.**    The major contributions of our work can be summarized as follows:

1. We propose a constrained optimization framework centered around a versatile family of norm constraints, which, together with the SFW algorithm, can result in well-performing models that are robust towards convolutional filter pruning as well as low-rank matrix decomposition. We empirically show on benchmark image-classification architectures and datasets that the proposed method is able to perform on par to or better than existing approaches. Especially in the case of low-rank decomposition, our approach can require much less computational resources than nuclear-norm regularization based approaches.

2. As a special case, our derivation includes a setting suitable for unstructured pruning. We show that our approach enjoys favorable properties when compared to the existing $k$-sparse approach (Pokutta et al., 2020; Miao et al., 2022), which we improve upon.

3. We empirically show that the robustness of SFW can largely be attributed to the usage of the *gradient rescaling* of the learning rate, which increases the batch gradient norm and effective learning rate throughout training, even though the train loss constantly decreases. To justify the usage of gradient rescaling theoretically, we prove the convergence of SFW with batch gradient dependent step size in the non-convex setting.

Compression-aware training is a promising research direction for training models to state-of-the-art performance while encouraging stability to pruning. We believe that our work is an important building block in the design of structured training algorithms. One strength is the fact that the proposed methods cover a wide range of compression domains, i.e., structured pruning, matrix decomposition as well as unstructured pruning. Our results show the suitability of the SFW algorithm and highlight the importance of the learning rate rescaling, which we justify theoretically in the hope of enabling further research.

**Related Work.** The *Frank-Wolfe* (FW) (Frank et al., 1956) or *conditional gradient* (Levitin & Polyak, 1966) algorithm has been studied extensively in the convex setting, enjoying popularity throughout various domains of Machine Learning for being able to efficiently deal with complex structural requirements (e.g. Lacoste-Julien et al., 2013; Zeng & Figueiredo, 2014; Frandi et al., 2015; Jaggi, 2013; Négiar et al., 2020). With convexity being a relatively strong assumption, Lacoste-Julien (2016) extended the convergence theory of FW to the non-convex setting, while Hazan & Luo (2016) and Reddi et al. (2016) provide convergence rates for the stochastic variant of the algorithm (SFW). Several different accelerated variants have been proposed, including variance reduction methods (Hazan & Luo, 2016; Yurtsever et al., 2019; Shen et al., 2019), adaptive gradients (Combettes et al., 2020) and momentum (Mokhtari et al., 2018; Chen et al., 2018).

With the theoretical foundation for the non-convex stochastic setting being laid, SFW has received a surge of interest in the context of training Neural Networks: Ravi et al. (2018) advocate the usage of parameter constraints in Deep Learning, Xie et al. (2019) train shallow networks using SFW, Berrada et al. (2018) design a variant specifically for Neural Networks, Tsiligkaridis & Roberts (2020) employ SFW for adversarial training and Pokutta et al. (2020) show that SFW can reach state-of-the-art performance on benchmark image classification tasks. While there is a rich literature on classical FW being applied to sparsity problems in Machine Learning, only few have considered exploiting the structure-enhancing properties of (stochastic) FW in the field of Deep Learning. Grigas et al. (2019) remove neurons from three layer convolutional architectures. Pokutta et al. (2020) propose to constrain the parameters to lie within a $k$-sparse polytope, resulting in a large fraction of the parameters having small magnitude. Miao et al. (2022) leverage this idea in the context of unstructured magnitude pruning with a focus on pruning-aware training, being a compression setting of training 'once-for-all' sparsities (Cai et al., 2020). For a detailed account of different sparsification approaches we refer to the excellent survey of Hoefler et al. (2021).

**Outline.** We begin by introducing the problem setting and the SFW algorithm. Section 3 contains a precise description of the proposed approach and Section 4 is devoted to experimentally comparing it to existing approaches. Section 5 contains an analysis of the two learning rate rescaling mechanisms and the converge theorem for gradient rescaling. Finally, we conclude and discuss the findings of our work in Section 6.

## 2 PRELIMINARIES

For $x \in \mathbb{R}^n$, we denote the $i$-th coordinate of $x$ by $[x]_i$. The diagonal matrix with $x$ on its diagonal is denoted by $\text{diag}(x) \in \mathbb{R}^{n,n}$. For $p \in [1, \infty]$, the $L_p$-ball of radius $\tau$ is denoted by $B_p(\tau)$. $\|x\|_0$ denotes the number of non-zero components of $x \in \mathbb{R}^n$. For any compact convex set $\mathcal{C} \subseteq \mathbb{R}^n$, let us further denote the $L_2$-diameter of $\mathcal{C}$ by $\mathcal{D}(\mathcal{C}) = \max_{x,y \in \mathcal{C}} \|x - y\|_2$. As usual, we denote the gradient of a function $L$ at $\theta$ by $\nabla L(\theta)$ and the batch gradient estimator by $\widetilde{\nabla}\mathcal{L}(\theta)$. For the sake of convenience, we abuse notation and apply univariate functions to vectors in an elementwise fashion, e.g., $|x|$ denotes the vector $|x| := (|x_1| \ldots, |x_n|)$. If not indicated otherwise, we treat a tensor $x$ of a network as a vector $x \in \mathbb{R}^n$.

**Constrained optimization using the Stochastic Frank-Wolfe algorithm** We aim at optimizing the parameters $\theta$ of a Neural Network while enforcing structure-inducing constraints by considering the constrained finite-sum optimization problem

$$\min_{\theta \in \mathcal{C}} L(\theta) = \min_{\theta \in \mathcal{C}} \frac{1}{m} \sum_{i=1}^{m} \ell_i(\theta), \tag{1}$$

where the per-sample loss functions $\ell_i$ are differentiable in $\theta$ and $\mathcal{C}$ is a compact, convex set. When using SGD, imposing hard constraints requires a potentially costly projection back to $\mathcal{C}$ to ensure feasiblity of the iterates. However, an alternative is the *Stochastic Frank-Wolfe* (SFW) algorithm (Frank et al., 1956; Berrada et al., 2018; Pokutta et al., 2020), being projection-free and perfectly suited for yielding solutions with structural properties. To ensure feasibility of the iterates, SFW does not use the (batch) gradient direction for its updates but rather chooses a boundary point or vertex of $\mathcal{C}$ that is best aligned with the (negative) gradient. In each iteration $t$, SFW calls a *linear minimization oracle* (LMO) on the stochastic batch gradient $\nabla_t = \widetilde{\nabla}\mathcal{L}(\theta_t)$ to solve

$$v_t = \arg\min_{v \in \mathcal{C}} \langle v, \nabla_t \rangle, \tag{2}$$

which is then used as the direction to update the parameters using the convex combination

$$\theta_{t+1} \leftarrow (1 - \eta_t)\theta_t + \eta_t v_t, \tag{3}$$

where $\eta_t \in [0, 1]$ is a suitable learning rate. If the initial parameters $\theta_0$ are ensured to lie in the convex set $\mathcal{C}$, then the convex update rule ensures feasibility of the parameters in each iteration. Solving Equation (2) is often much cheaper than performing a projection step (Jaggi, 2013; Combettes & Pokutta, 2021), in many cases even admitting a closed-form solution. If $\mathcal{C}$ is given by the convex hull of (possibly infinitely many) vertices, a so-called *atomic domain*, then the solution to Equation (2) is attained at one of these vertices (Jaggi, 2013).

**Inducing structure through the feasible region**   Apart from constraining the parameters of a network to satisfy a certain norm constraint, say to have bounded euclidean norm as is typically done with weight decay (being equivalent via the Lagrangian formulation), the unique update rule Equation (3) of the SFW algorithm can be used to induce structure through the feasible region. Not only can a feasible region where the $v_t$ are highly structured be beneficial to generalization (Ravi et al., 2018; Pokutta et al., 2020), but further induce desirable properties such as sparsity to the network itself.

A recent example is the *k-sparse polytope* introduced by Pokutta et al. (2020), being a generalization of the $L_1$-ball $B_1$. The $k$-sparse polytope $\mathcal{C} = P_k(\tau)$ is defined as the convex hull of all vectors $v \in \{0, \pm\tau\}^n$ with at most $k$ non-zero entries, which per design, form the solution set to Equation (2). For small $k$, $v_t$ exhibits a high degree of sparsity and by Equation (3) only $k$ parameters are activated while all remaining parameters are discounted strongly, encouraging convergence towards sparse solutions (Pokutta et al., 2020; Miao et al., 2022).

## 3   METHODOLOGY: COMPRESSION-AWARE TRAINING WITH SFW

In the following, we propose leveraging a suitable family of norm constraints which arise naturally from $L_2$-regularization with sparsification requirements. In general, we require the constraints to result in sparse update directions when applying the update rule of Equation (3). In particular, we want to discriminate between predefined groups of parameters, that is, we aim at decaying seemingly unimportant groups of parameters (e.g. filters or neurons) while allowing others to grow. Similarly to the proposals by Pokutta et al. (2020) and Miao et al. (2022), we control the degree of sparsification with a tunable hyperparameter $k$ such that the update vectors $v_t$ are $k$-sparse, i.e., non-zero at at most $k$ entries. However, the existing approach is limited to the sparsification of Neural Networks on an individual-weight basis (i.e. unstructured pruning) and may further lead to hyperparameter and convergence instability, which we discuss and improve upon with our proposal.

Distinguishing this work from previously proposed approaches, we aim at constructing constraints for the structured sparsification case, which elegantly include the unstructured case as well. In addition, we ensure that (similar to classical SGD), the individual parameters receive updates corresponding to the magnitude of the gradient, enabling better convergence independent of the actual $k$ chosen, as we will discuss further below.

### 3.1   INDUCING GROUP SPARSITY TO NEURAL NETWORKS

Given a disjoint partition of the network's parameters into groups $G \in \mathcal{G}$, we define the *group-k-support norm* (Rao et al., 2017) ball of radius $\tau$ as

$$\mathcal{C}_k^{\mathcal{G}}(\tau) = \text{conv}\{v \mid \|v\|_{0,\mathcal{G}} \leq k, \|v\|_2 \leq \tau\}, \tag{4}$$

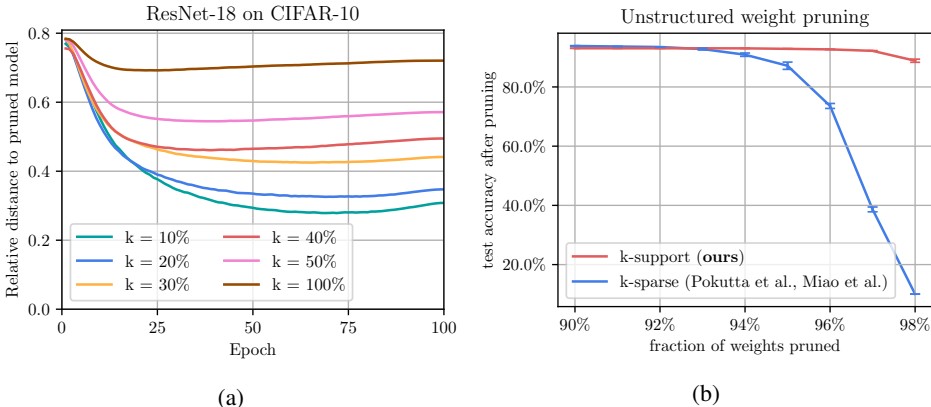

(a)                                    (b)

Figure 1: ResNet-18 on CIFAR-10: Relative distance to filter pruned model corresponding to 70% sparsity when training with the proposed approach and varying $k$ (left) and accuracy-vs.-sparsity tradeoff curves for unstructured weight pruning comparing our approach to the existing $k$-sparse approach (right).

where $\|v\|_{0,\mathcal{G}}$ is the smallest number of groups that are needed to cover the support of $v$. In other words, the vertex set of $\mathcal{C}_k^{\mathcal{G}}(\tau)$ is given by all (vectorized) parameters for which the euclidean norm is bounded by $\tau$ and where at most $k$ groups contain non-zero entries. Here, the definition of a set of groups $\mathcal{G}$ is left abstract, as it could be any disjoint partition of the parameters, that is each parameter $w$ must lie in exactly one group $w \in G \in \mathcal{G}$. In the following however, let $\mathcal{G}$ be the set of all filters in a convolutional layer.

The norm was originally proposed by Rao et al. (2017) motivated by the group lasso (Yuan & Lin, 2006), which is common in the statistics and classical machine learning literature. Choosing $\mathcal{G}$ as a partition of the networks parameters, we can now state the solution to Equation (2) as follows. For a group $G \in \mathcal{G}$ let $\|x\|_G$ be the $L_2$-norm of $x \in \mathbb{R}^n$ when only considering elements of $G$. Given the batch gradient of the $l$-th convolutional layer $\nabla_t^l$, let $G_1, \ldots, G_k$ be the $k$ groups with the largest gradient $L_2$-norm $\|\nabla_t^l\|_{G_i}$ and let $H = \bigcup_{i=1}^k G_i$. The solution to Equation (2) is then given by

$$[v_t]_i = \begin{cases} -\tau [\nabla_t^l]_i / \|\nabla_t^l\|_H & \text{if } i \in H, \\ 0 & \text{otherwise.} \end{cases}$$

A proof of this fact can be found in Rao et al. (2017). SFW applied to $\mathcal{C}_k^{\mathcal{G}}(\tau)$ updates the $k$ filters whose (stochastic) gradient entries correspond to those of fastest loss decrease while accounting for the distribution of magnitude among them instead of using the same magnitude for all parameters. Due to the convex update rule, the remaining filters are decayed, eventually resulting in few of them not being close to zero and thus making the trained network robust towards filter pruning. Figure 1a shows how different values of $k$ as a fraction of the overall number of filters influence the relative distance to the pruned model, indicating that $k$ allows controlling the degree of robustness towards sparsification.

**Unstructured sparsity as a special case** Clearly, the proposed approach is easily extendable to the pruning of other groups, such as neurons. A most obvious special case arises when each weight of the network has its own group, naturally extending the above rationale to the unstructured pruning case. In that case, one recovers the *k-support norm* as proposed by Argyriou et al. (2012), which is a suitable candidate for encouraging robustness towards unstructured weight pruning and to comparison to the existing $k$-sparse polytope approach leveraged by Miao et al. (2022).

The $k$-sparse approach approach suffers from two drawbacks, which are mitigated by our proposed method. First of all, all activated parameters will receive an update of same magnitude, namely $\tau$. This hinders convergence, especially when $k$ is larger. Consider for example the worst-case scenario in which $k$ equals the number of parameters $n$, then every single parameter of the network will receive an update of magnitude $\tau$, essentially losing the entire information of the gradient apart from the

entrywise sign. On the contrary, the $k$-support norm with $k = n$ will lead to optimization over the $L_2$-ball, yielding the default and best converging case of Neural Network training. Figure 1b shows the facilitated convergence of our approach, which is nonetheless highly robust towards unstructured pruning. The $k$-sparse approach performs well in the medium to high sparsity regime, but quickly collapses for higher compression rates. A clear advantage of $k$-support norm ball constraints is that SFW is able to obtain this performance in the high compression regime while not suffering from underperformance before pruning. Secondly, Pokutta et al. (2020) and Miao et al. (2022) specify the desired $L_2$-diameter $\mathcal{D}$ of $\mathcal{C} = P_k(\tau)$ to control the regularization strength and then in turn choose the radius $\tau$ such that $\mathcal{D}(P_k(\tau)) = \mathcal{D}$. Defined this way, $\tau$ depends on $k$ as $\tau = \mathcal{D}/(2\sqrt{k})$. This is counter-intuitive, since $k$ controls both the amount of activated parameters as per design of the LMO as well as the magnitude of the parameter updates, resulting in unnecessarily coupled parameters. As opposed to the $k$-sparse polytope, the diameter of the $k$-support norm ball does not depend on $k$ and hence decouples the parameters $k$ and $\tau$ as desired. Figure 4 in the appendix shows the successful decoupling of the radius and $k$. The $k$-support norm ball is less sensitive to hyperparameter changes and obtains better results throughout a wide range of hyperparameter configurations than its $k$-sparse counterpart.

## 3.2 A DIFFERENT NOTION OF SPARSITY: PRUNING SINGULAR VALUES

So far we employed a very restricted notion of *sparsity*, namely that of the existence of zeros in a matrix or tensor. Instead of removing individual parameters or groups thereof, networks can also be compressed after training by decomposing parameter matrices into a product of smaller matrices, allowing one to replace a layer by two consecutive ones that require a drastically smaller amount of FLOPs at inference (Denton et al., 2014). The key ingredient is the truncated singular value decomposition (SVD), where setting the smallest singular values to zero leads to an optimal low-rank approximation by virtue of the Eckart–Young–Mirsky theorem. More precisely, given a rank $r$ parameter matrix $\mathcal{W} \in \mathbb{R}^{n,m}$ with singular values $\sigma(\mathcal{W}) = (\sigma_1(\mathcal{W}), \ldots, \sigma_r(\mathcal{W}))$ and SVD $\mathcal{W} = \sum_{i=1}^{r} u_i \sigma_i v_i^T = U\Sigma V^T$, $U = [u_1, \ldots, u_r] \in \mathbb{R}^{n,r}$, $\Sigma = \text{diag}(\sigma(\mathcal{W})) \in \mathbb{R}^{r,r}$, $V = [v_1, \ldots, v_r] \in \mathbb{R}^{m,r}$, the $k$-SVD of $\mathcal{W}$ is given by $\mathcal{W} \approx \sum_{i=1}^{k} u_i \sigma_i v_i^T = U_k \Sigma_k V_k^T$, where the magnitude of 'pruned' singular values quantifies the error in approximation. A detailed account of this approach can be found in the appendix. A natural approach to ensure robustness to matrix decomposition is hence based on penalizing the nuclear norm $\|\mathcal{W}\|_* := \|\sigma(\mathcal{W})\|_1$ (Tai et al., 2015; Alvarez & Salzmann, 2017), which requires the costly computation of the full SVD in each iteration.

When constraining the parameters to have bounded nuclear norm instead of penalizing it, the LMO solution to Equation (2) utilized by SFW can be computed efficiently by requiring only the first singular value-vector-pair (Jaggi, 2013). Extending this notion to consider the $k$ largest singular pairs, we propose utilizing the *spectral-$k$-support norm* (McDonald et al., 2016), for which the ball of radius $\tau$ is defined as

$$\mathcal{C}_k^\sigma(\tau) = \text{conv}\{\mathcal{W} \in \mathbb{R}^{n \times m} \mid \text{rank}(\mathcal{W}) \leq k, \|\sigma(\mathcal{W})\|_2 \leq \tau\}, \tag{5}$$

where $\|\sigma(\mathcal{W})\|_2$ is the 2-Schattennorm of the singular values $\sigma(\mathcal{W})$ (Jaggi, 2013). The following lemma allows us to efficiently compute the LMO solution.

**Lemma 3.1.** *Given $\nabla_t \in \mathbb{R}^{n \times m}$, let $\mathcal{W}_t = -\tau\|\sigma(\Sigma_k)\|_2^{-1} U_k \Sigma_k V_k^T \in \mathcal{C}_k^\sigma(\tau)$, where $U_k \Sigma_k V_k^T$ is the truncated $k$-SVD of $\nabla_t$ such that only the $k$ largest singular values are kept. Then $\mathcal{W}_t$ is a solution to Equation (2).*

A proof can be found in the appendix. Note that similar to the group-$k$-support norm ball taking the magnitude of $k$ largest gradient groups into account, the scaling by $\Sigma_k \|\sigma(\Sigma_k)\|_2^{-1}$ takes the magnitude of $k$ largest singular values into account. In Section 4.2 we study the capabilities of SFW when constraining the Spectral-$k$-support norm of convolutional tensors, which account for the majority of FLOPs at inference (Han et al., 2015). While there exist higher-order generalizations of the SVD to decompose tensors directly (cf. Lebedev et al., 2014; Kim et al., 2015), we follow the approach of interpreting the tensor $\mathcal{W} \in \mathbb{R}^{n \times c \times d \times d}$ with $c$ in-channels, $n$ convolutional filters and spatial size $d$ as an $(n \times cd^2)$-matrix (Alvarez & Salzmann, 2017; Idelbayev & Carreira-Perpinán, 2020).

### 3.3 EXPERIMENTAL SETUP: A FAIR COMPARISON BETWEEN METHODS

All experiments are conducted using the *PyTorch* framework (Paszke et al., 2019), where we relied on *Weights & Biases* (Biewald, 2020) for the analysis of results. To enable reproducibility, our implementation will be available throughout the entire review process and publicly thereafter.

We train convolutional architectures such as *Residual Networks* (He et al., 2015) and *Wide Residual Networks* (Zagoruyko & Komodakis, 2016) on *ImageNet-1K* (Russakovsky et al., 2015), *TinyImageNet* (Le & Yang, 2015), *CIFAR-100* and *CIFAR-10* (Krizhevsky et al., 2009). The exact training setups as well as grid searches used can be found in the appendix. As a general remark, we follow the experimental guidelines of Blalock et al. (2020) towards standardized comparisons between sparsification methods. All results are averaged over two seeds with min-max bands indicated for plots and standard deviation for tables. We use a validation set of 10% of the training data for hyperparameter selection.

In the compression-aware setting we are interested in finding single hyperparameter configurations that perform well under a wide variety of compression rates, i.e., without tuning hyperparameters for each sparsity. When comparing the performance for multiple compression rates at once, we have to decide how to select the 'best' hyperparameter configuration. To that end, we select the configuration for each method that results in the highest on-average validation accuracy among all sparsities at stake.

## 4 EXPERIMENTAL RESULTS

### 4.1 STRUCTURED FILTER PRUNING

For the pruning of convolutional filters, we follow the *PFEC* approach of Li et al. (2016): at the end of training we sort the filters of each convolutional layer by their $L_1$-norm and remove the smallest ones until the desired level of compression is met. We enforce a uniform distribution of sparsity among layers, noting that there exist more sophisticated ways of determining the per-layer pruning ratios (Liebenwein et al., 2021).

**Baselines** We compare our approach (denoted as *SparseFW*) to several baselines. Every experiment includes the most natural baseline, which corresponds to regular training with momentum SGD and weight decay. Apart from that, we have implemented the following recent filter pruning approaches. *SSL* (Wen et al., 2016) leverages a group penalty on the filters. Similarly, *GLT* (Alvarez & Salzmann, 2016) employs a group-lasso on the filters followed by a proximal gradient descent (soft-thresholding) step. *ABFP* (Ding et al., 2018) follows an "auto-balanced" approach which penalizes certain filters while actively encouraging others to grow. *SFP* (He et al., 2018) softly prunes after each epoch, allowing filters to recover throughout the epoch.

**Results** Figure 2 shows that SFW converges to solutions that are robust to a wide range of filter pruning ratios. Especially in the high sparsity regime SFW is able to keep most of its performance, while other approaches collapse, with the exception of ABFP for TinyImageNet which can be even more robust for high sparsities. Apart from that, SFW reaches excellent results for a wide range of compression ratios. Full results can be found in the appendix.

### 4.2 LOW-RANK MATRIX DECOMPOSITION

We compare SparseFW with spectral-$k$-support norm constraints to other approaches aiming for robustness to tensor decomposition. At the end of training, we set the smallest singular values of each convolutional matrix to zero and replace the layer by two consecutive layers as described in the appendix.

**Baselines** Apart from the regular training baseline using momentum SGD with weight decay, we implemented NUC (Denton et al., 2014) and SVDEnergy (Alvarez & Salzmann, 2017). The former employs a nuclear norm regularization technique together with SGD, hence computing the subgradient of a nuclear norm regularization penalty term (Watson, 1992) and updating the weights accordingly. The latter similarly performs the usual SGD update on the loss, followed by applying

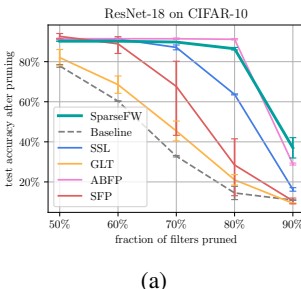 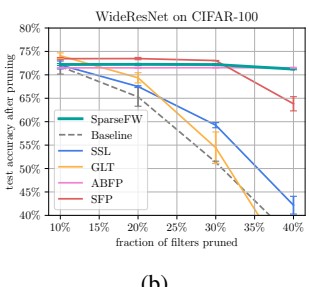 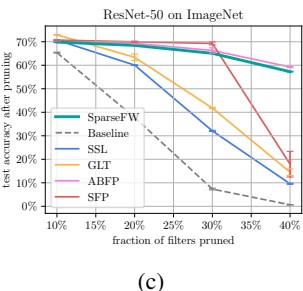

| (a) | (b) | (c) |

Figure 2: Accuracy-vs.-sparsity tradeoff curves for structured convolutional filter pruning on CIFAR-10 (a), CIFAR-100 (b) and ImageNet (c). The plots show the parameter configuration with highest test accuracy after pruning when averaging over all sparsities at stake.

Table 1: WideResNet on CIFAR-100: Comparison of approaches for encouraging low-rank matrices throughout training.

| | **Sparsity** | | | | | |
|---|---|---|---|---|---|---|
| **Method** | 30 % | 40% | 50% | 60% | 70% | 80% |
| Baseline | 75.57 ±0.10 | 74.28 ±2.59 | 73.52 ±0.95 | 64.67 ±11.12 | 56.04 ±2.24 | 10.97 ±8.29 |
| **SparseFW** | 75.53 ±0.18 | 75.69 ±0.04 | 75.75 ±0.08 | 75.37 ±0.40 | 75.30 ±0.10 | 73.28 ±2.02 |
| **NUC** | 75.96 ±0.29 | 74.86 ±1.53 | 72.35 ±0.48 | 68.57 ±4.95 | 55.97 ±2.80 | 5.40 ±2.12 |
| **SVDEnergy** | 75.69 ±0.95 | 75.14 ±0.18 | 74.11 ±0.81 | 66.76 ±4.41 | 55.40 ±3.04 | 30.82 ±26.87 |

the soft-thresholding operator over the singular values of the parameters, a strategy also known as *singular value thresholding* (Cai et al., 2010). Both approaches require the computation of the entire SVD, tradionally requiring $\mathcal{O}(nm \min(n, m))$ operations for an $n \times m$ matrix (Allen-Zhu & Li, 2016).

**Results** Table 1 compares the post-pruning test accuracy of the four approaches for a range of sparsities between 30 and 80 percent. The proposed approach outperforms both the natural baseline of regular SGD training as well as the nuclear norm regularization approaches, experiencing only a minor accuracy decrease in the high sparsity regime and no performance degradation medium sparsity regime. The accuracies correspond to the on-average best hyperparameter configuration. A significant advantage of SFW in this regard is its efficiency, allowing a higher images-per-second throughput as nuclear norm regularized based approaches (446 images per second compared to 191 images per second on CIFAR-100), where we performed measurements on the same hardware, namely a 24-core Xeon Gold with Nvidia Tesla V100 GPU. Since the LMO requires the computation of the $k$ largest singular pairs, the efficiency of SFW is clearly dependent on $k$. The on-average best configuration of SFW in this setting is given by $k$ corresponding to ten percent of the singular values. We used a naive implementation of the $k$-SVD power method (Bentbib & Kanber, 2015), noting that there are more sophisticated and faster algorithms (Allen-Zhu & Li, 2016).

## 5 THE DYNAMICS OF GRADIENT RESCALING

The learning rate $\eta_t \in [0, 1]$ determines the length of the parameter update relative to the size of the feasible region. This coupling between regularization strength and step size makes the tuning of the learning rate cumbersome. To decouple the tuning of the learning rate from the size of the feasible region, Pokutta et al. (2020) propose two different learning rate rescaling mechanisms: *diameter rescaling* and *gradient rescaling*, the latter being used throughout our experiments in the preceding section. While the former divides the learning rate by the $L_2$-diameter $\mathcal{D}(\mathcal{C})$ of $\mathcal{C}$, gradient rescaling rescales the update direction length to that of the batch gradient, i.e., $\hat{\eta}_t := \eta_t \|\nabla_t\|_2 / \|v_t - \theta_t\|_2$.

It is however largely unclear what effect these normalization schemes have on both the convergence for regular training as well as the robustness to pruning, noting that the learning rate of SFW explicitly

controls the decay on non-activated parameters. Figure 3 shows the test accuracy vs. sparsity tradeoff curves directly before and after magnitude-pruning of the parameters, comparing the two rescaling variants when training with $k$-support norm constraints. Gradient rescaling consistently outperforms its diameter-based counterpart w.r.t. both dense as well as pruned test accuracy.

We found the denominator of gradient rescaling not to be subject to much variation, whereas the batch gradient norm dynamically changes the learning rate over time. Figure 6 compares the evolution of $\|\nabla_t\|$ for two different radii of the $k$-support norm ball (with fixed $k$), where we compare to usual SGD training with weight decay. For both SGD and SFW, $\|\nabla_t\|$ is subject to noise and increases until 75% of the training process, despite the continuous decrease of the train loss. In fact, the batch gradient norm is not significantly smaller than at the start of training even though the loss converges. This behaviour might best be explainable by the presence of *Batch-Normalization* layers, whose interplay with weight decay has recently been analyzed by van Laarhoven (2017) and Hoffer et al. (2018): layers preceding a Batch-Normalization are rescaling invariant, that is their output remains unchanged when multiplying all parameters by a scalar, however rescaling them results in inverse rescaling of the gradient norm in subsequent layers and iterations. Weight decay continuously decreases the scale of the parameters and hence increases the scale of the batch gradient, where stronger decay of the former

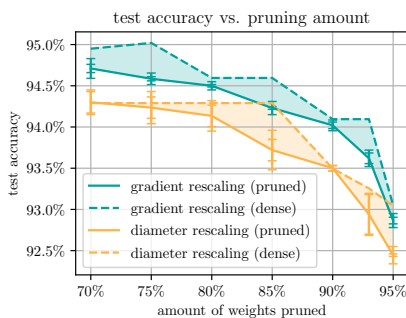

Figure 3: ResNet-18 on CIFAR-10: For each pruning amount, the best hyperparameter configuration w.r.t. the accuracy after pruning (*pruned*) is depicted. The corresponding value before pruning (*dense*) is depicted as a dashed line.

leads to stronger increase of the latter. Since in gradient rescaling the norm of the batch gradient also influences the strength of the decay of the parameters, this process has a self-accelerating dynamic. This dynamic results in larger steps towards the (sparse) vertices of the $k$-support norm ball, leading to a stronger decay on the previous parameter configuration, which in turn increases the robustness to pruning, making gradient rescaling the method of choice in that setting.

With the following theorem, we lay the theoretical foundation by showing that incorporating the batch gradient norm into the learning rate leads to convergence of SFW at the specified rate, that is, the expected product of exact gradient norm and the *Frank-Wolfe Gap* $\mathcal{G}(\theta_t)$ decays at a rate of $\mathcal{O}(T^{-1/2})$. The precise statement as well as a proof can be found in Appendix A.5.

**Theorem 5.1** (Convergence of gradient rescaling, informal). *Assume that $L$ is M-smooth and $\ell$ is G-Lipschitz and let $\eta_t = \|\nabla_t\|\eta$ for appropriately chosen $\eta$ and all $0 \leq t < T$. If $\theta_a$ is chosen uniformly at random from the SFW iterates $\{\theta_i : 0 \leq i < T\}$, then we have $\mathbb{E}\left[\mathcal{G}(\theta_a) \cdot \|\nabla L(\theta_a)\|\right] = \mathcal{O}(T^{-1/2})$, where $\mathbb{E}$ denotes the expectation w.r.t. all the randomness present.*

## 6 DISCUSSION AND OUTLOOK

We proposed to utilize a versatile family of norm constraints to, together with the SFW algorithm, train deep neural networks to state-of-the-art dense performance as well as robustness to compression for a wide range of compression ratios. Our experimental results show that SFW can leverage highly structured feasible regions to avoid performance degradation when performing convolutional filter pruning or low-rank tensor decomposition. For the latter, SFW can result in significant speedups compared to nuclear-norm regularization based approaches. As a special case, our proposed approach includes the unstructured pruning case and we showed how utilizing the proposed norm can mitigate the drawbacks of and improve upon the results of Miao et al. (2022). We hope that our findings regarding the importance of the learning rate rescaling as well as Theorem 5.1 stimulate further research in the direction of compression-aware training with SFW.

However, we emphasize that our results hold primarily in the setting that we described, namely that of compression-aware training, where the training is sparsity-agnostic and retraining is prohibitive. Our goal was to show the versatility of SFW, which provides a suitable algorithmic framework for enforcing structure throughout training. If the sparsity can be incorporated into training, significantly more complex approaches can be applied.

## 7 REPRODUCIBILITY

Reproducibility is a crucial aspect of any work based on experiments. Our code will be available to the reviewers throughout all stages of the submission and publicly thereafter. We also highlight some publicly available implementations we used. The ResNet-18 implementation is based on `https://github.com/charlieokonomiyaki/pytorch-resnet18-cifar10/blob/master/models/resnet.py`, where for the WideResNet architecture we relied on `https://github.com/meliketoy/wide-resnet.pytorch`. For the computation of the theoretical speedup, we used the implementation of Blalock et al. (2020) available at `https://github.com/JJGO/shrinkbench`. Regarding pruning methods that are not directly available through the Pytorch framework, we used the original implementation whenever possible and indicate so directly in the code.

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

# A    APPENDIX

## A.1    TECHNICAL DETAILS

Whenever using SFW, we employ a momentum variant (Pokutta et al., 2020) and enforce *local* constraints, i.e., we constrain the parameters of each layer in a separate feasible region. As suggested by Pokutta et al. (2020), we do not tune the radius $\tau$ of the feasible region $\mathcal{C}(\tau)$ directly but rather specify a scalar factor $w > 0$ and set the $L_2$-diameter of the feasible region of each layer as

$$\mathcal{D} = 2w\mathbb{E}(\|\theta\|_2),$$

where the expected $L_2$-norm of the layer parameters $\theta$ are simply estimated by computing the mean among multiple default initializations. $\mathcal{D}$ is then in turn used to compute $\tau$ such that $\mathcal{D}(\mathcal{C}(\tau)) = \mathcal{D}$. This allows us to control the $L_2$-regularization strength of each layer. We do not prune biases and batch-normalization parameters, as they only account for a small fraction of the total number of parameters yet are crucial for obtaining well-performing models (Evci et al., 2020). Apart from the direct comparison between diameter and gradient rescaling, we use gradient rescaling throughout all experiments.

While there exist multiple successful strategies to retrain after pruning (Renda et al., 2020; Le & Hua, 2021; Zimmer et al., 2021), the compression-aware training setting requires the methods to be compared directly after pruning without retraining. However, similarly to Li et al. (2020a) and Peste et al. (2022) we notice that the validation accuracy after pruning can be significantly increased by recomputing the Batch Normalization (Ioffe & Szegedy, 2015) statistics, which have to be recalibrated since the pre-activations of the hidden layers are distorted by pruning. To that end, we recompute the statistics on the train dataset after pruning and note that in practice only a fraction of the training data is necessary to recalibrate the Batch Normalization layers.

In the following, we state technical details of our approaches to structured and unstructured pruning, as well as low-rank matrix decomposition.

### A.1.1    STRUCTURED FILTER AND UNSTRUCTURED PRUNING

As outlined in the main section, we follow the approach of Li et al. (2016) and remove the convolutional filters with smallest $L_1$-norm. Since each filter might correspond to a different number of parameters, depending on the convolutional layer it is located in, the $L_1$-norm of filters is incomparable among different layers. We hence follow the local approach and prune the same amount in each convolutional layer until the desired sparsity is met. Since this will lead to the same theoretical speedup, independent of the algorithm at stake, we omit these values.

For unstructured pruning, we employ the usual magnitude pruning, that is, we remove the parameters with smallest absolute value until we meet the desired level of compression. As we found it to work best among all algorithms, we prune *globally*, i.e. we select the smallest weights among all network parameters eligible for pruning. We note however that there exists several different magnitude-based selection approaches (cf. e.g. Han et al., 2015; Gale et al., 2019; Lee et al., 2020).

### A.1.2    LOW-RANK MATRIX DECOMPOSITION

We describe the rationale behind the decomposition of matrices with (preferably) low rank. Low-Rank matrix decomposition is centered around the idea of truncating the singular value decomposition (SVD). Let $\theta = U\Sigma V^T = \sum_{i=1}^{r} u_i\sigma_i v_i^T$ be the SVD of rank-$r$ matrix $\theta \in \mathbb{R}^{n \times m}$, where $u_i, v_i$ are the singular vectors to singular values $\sigma_1 \geq \ldots \geq \sigma_r > 0$. The goal is now to find the best low-rank approximation $\hat{\theta} \in \mathbb{R}^{n \times m}$ to $\theta$, namely to solve

$$\min_{\text{rank}(\hat{\theta}) \leq t} \|\theta - \hat{\theta}\|_F. \tag{6}$$

By the Eckart–Young–Mirsky theorem, a minimizer is given by $U_t\Sigma_t V_t^T := \sum_{i=1}^{t} u_i\sigma_i v_i^T$, where the smallest $t - r$ singular values are truncated. If the singular values now decay rapidly, i.e., only the first $t$ singular values contain most of the *energy* of $\theta$, then this approximation can be applied as a post-processing step without much change in the output of a linear layer (Denton et al., 2014). Consequently, a natural approach to ensure stability w.r.t. matrix decomposition is based on

Table 2: Exact training configurations used throughout the experiments. The dense test accuracy refers to the optimal accuracy we achieve using momentum SGD with weight decay.

| Dataset | Network (number of weights) | Epochs | Batch size | Momentum | Learning rate ($t$ = iteration) | Dense test accuracy |
|---|---|---|---|---|---|---|
| CIFAR-10 | ResNet-18 (11 Mio) | 100 | 128 | 0.9 | linear decay from 0.1 | 95.0% ±0.04% |
| CIFAR-100 | WideResNet-28x10 (37 Mio) | 100 | 128 | 0.9 | linear decay from 0.1 | 76.7% ±0.2% |
| TinyImagenet | ResNet-50 (26 Mio) | 100 | 128 | 0.9 | linear decay from 0.1 | 64.9% ±0.1% |
| ImageNet | ResNet-50 (26 Mio) | 90 | 1024 | 0.9 | linear decay from 0.1 | 75.35% ±0.1% |

encouraging the parameter matrices to have low-rank throughout training, most prominently done by regularizing the nuclear norm of the weight matrix (Tai et al., 2015; Alvarez & Salzmann, 2017). If $t$ is small, then the number of FLOPs can be drastically reduced by decomposing a layer into multiple layers. For linear layers this is straightforward: let $(\theta, \beta)$ be weights and biases of a linear layer with SVD of $\theta$ as above. For input $x$, the layer computes

$$\theta x + \beta = U\Sigma V^T x + \beta \approx U_t(\Sigma_t V_t^T x) + \beta, \tag{7}$$

being interpretable as the consecutive application of two linear layers: $(\Sigma_t V_t^T, 0)$ followed by $(U_t, \beta)$, possibly reducing the number of parameters from $nm$ to $t(n + m)$. For a four-dimensional convolutional tensor $\theta \in \mathbb{R}^{n \times c \times d \times d}$, where $c$ is the number of in-channels, $n$ the number of convolutional filters, and $d$ is the spatial size, we cannot directly construct the SVD. However, we follow an approach similar to those of Alvarez & Salzmann (2017) and Idelbayev & Carreira-Perpinán (2020), interpreting $\theta$ as a $(n \times cd^2)$ matrix, whose truncated SVD decomposition allows us to replace the layer by two consecutive convolutional layers, the first one having $t$ filters, $c$ channels and spatial size $d$, followed by $n$ filters, $t$ channels and spatial size of one.

We describe NUC and SVDEnergy in more detail. Both approaches solve the problem

$$\min_\theta L(\theta) + \lambda \|\theta\|_*, \tag{8}$$

where $\lambda > 0$ is a suitably chosen regularization parameter. NUC does so by computing the subgradient of $\lambda \|\theta\|_*$ (Watson, 1992) and performing an ordinary SGD update. On the other hand, SVDEnergy relies on the proximity operator of the nuclear norm, also known as *singular value thresholding* (Cai et al., 2010; Alvarez & Salzmann, 2017). In each iteration this amount to first applying an SGD step to minimize $L(\theta_t)$ with learning rate $\eta_t$ obtaining $\hat{\theta}_{t+1}$, which is then followed by soft-thresholding the singular values of $\hat{\theta}_{t+1}$ to obtain $\theta_{t+1}$, i.e.,

$$\theta_{t+1} = \sum_{i=1}^{\text{rank}(\hat{\theta}_{t+1})} u_i \max(0, \sigma_i - \eta_t \lambda) v_i^T, \tag{9}$$

where $u_i, v_i$ are the singular vectors of $\hat{\theta}_{t+1}$ to singular values $\sigma_i$. In essence, singular values smaller than $\eta_t \lambda$ are cut-off, potentially resulting in $\theta_{t+1}$ having low rank.

## A.2 EXPERIMENTAL SETUP AND EXTENDED RESULTS

Table 2 shows the exact training configurations we used throughout all experiments, where we relied on a training timeframe of 100 epochs with a linearly decaying learning rate, as suggested by Li et al. (2020b). In the following, we state the hyperparameter grids used as well as full tables and missing plots.

### A.2.1 STRUCTURED FILTER PRUNING

**CIFAR-10 Hyperparameter grids** If not specified otherwise, we use weight decay values of {1e-4, 5e-4} for all algorithms except SparseFW.

- SparseFW: We tune the fractional $k \in \{0.1, 0.2, 0.3\}$ and the multiplier $w \in \{10, 20, 30\}$ of the $L_2$-diameter.
- SSL: We tune the the filter group penalty factor $\lambda \in \{$1e-5, 5e-5, 1e-4, 5e-4, 1e-3, 5e-3$\}$.
- GLT: We tune the the filter group penalty factor $\lambda \in \{$1e-5, 5e-5, 1e-4, 5e-4, 1e-3, 5e-3$\}$ and the lasso tradeoff between 0 and 0.5.

- ABFP: We tune the fractional $k \in \{0.1, 0.2, 0.3, 0.4\}$ and the filter group penalty factor $\lambda \in \{$1e-5, 5e-5, 1e-4, 5e-4, 1e-3, 5e-3$\}$.

- SFP: We tune the fractional $k \in \{0.5, 0.6, 0.7, 0.8\}$. Further, we found it beneficial to tune the epoch at which SFP starts the sparsification between $\{0, 10, 25\}$, since to early starts might result in a model collapse.

**CIFAR-100 Hyperparameter grids**    If not specified otherwise, we use weight decay values of $\{$1e-4, 5e-4$\}$ for all algorithms except SparseFW.

- SparseFW: We tune the fractional $k \in \{0.15, 0.2, 0.25, 0.3\}$ and the multiplier $w \in \{20, 30, 40\}$ of the $L_2$-diameter.

- SSL: We tune the the filter group penalty factor $\lambda \in \{$1e-5, 5e-5, 1e-4, 5e-4, 1e-3, 5e-3$\}$.

- GLT: We tune the the filter group penalty factor $\lambda \in \{$1e-4, 5e-4, 1e-3, 5e-3, 1e-2, 5e-3$\}$ and the lasso tradeoff between $0$ and $0.5$.

- ABFP: We tune the fractional $k \in \{0.1, 0.2, 0.3\}$ and the filter group penalty factor $\lambda \in \{$1e-5, 5e-5, 1e-4, 5e-4, 1e-3, 5e-3$\}$.

- SFP: We tune the fractional $k \in \{0.6, 0.7, 0.8, 0.9\}$. Further, we found it beneficial to tune the epoch at which SFP starts the sparsification between $\{0, 10, 25\}$, since to early starts might result in a model collapse.

**TinyImagenet Hyperparameter grids**    If not specified otherwise, we use weight decay values of $\{$1e-4, 5e-4$\}$ for all algorithms except SparseFW.

- SparseFW: We tune the fractional $k \in \{0.15, 0.2, 0.25, 0.3\}$ and the multiplier $w \in \{20, 30, 40\}$ of the $L_2$-diameter.

- SSL: We tune the the filter group penalty factor $\lambda \in \{$1e-5, 5e-5, 1e-4, 5e-4, 1e-3, 5e-3$\}$.

- GLT: We tune the the filter group penalty factor $\lambda \in \{$1e-4, 5e-4, 1e-3, 5e-3, 1e-2, 5e-3$\}$ and the lasso tradeoff between $0$ and $0.5$.

- ABFP: We tune the fractional $k \in \{0.1, 0.2, 0.3\}$ and the filter group penalty factor $\lambda \in \{$1e-5, 5e-5, 1e-4, 5e-4, 1e-3, 5e-3$\}$.

- SFP: We tune the fractional $k \in \{0.6, 0.7, 0.75, 0.8, 0.85, 0.9\}$. Further, we found it beneficial to tune the epoch at which SFP starts the sparsification between $\{0, 20, 50\}$, since to early starts might result in a model collapse.

**Imagenet Hyperparameter grids**    If not specified otherwise, we use weight decay values of $\{$1e-4$\}$ for all algorithms except SparseFW.

- SparseFW: We tune the fractional $k \in \{0.2, 0.25, 0.3, 0.35\}$ and the multiplier $w \in \{20, 25, 30, 35\}$ of the $L_2$-diameter.

- SSL: We tune the the filter group penalty factor $\lambda \in \{$1e-5, 5e-5, 1e-4, 5e-4, 1e-3, 5e-3$\}$.

- GLT: We tune the the filter group penalty factor $\lambda \in \{$1e-4, 5e-4, 1e-3, 5e-3, 1e-2, 5e-3$\}$ and the lasso tradeoff between $0$ and $0.5$.

- ABFP: We tune the fractional $k \in \{0.1, 0.2, 0.3\}$ and the filter group penalty factor $\lambda \in \{$1e-5, 5e-5, 1e-4, 5e-4, 1e-3, 5e-3$\}$.

- SFP: We tune the fractional $k \in \{0.6, 0.7, 0.75, 0.8, 0.85, 0.9\}$. Further, we found it beneficial to tune the epoch at which SFP starts the sparsification between $\{0, 20, 50\}$, since to early starts might result in a model collapse.

Table 3: ResNet-18 on CIFAR-10: Comparison of Filter Pruning approaches. For each sparsity we indicate the achieved test accuracy after pruning averaged over all random seeds including standard deviation.

| | Sparsity | | | |
|---|---|---|---|---|
| **Method** | 60% | 70% | 80% | 90% |
| Baseline | 60.32 ±0.18 | 32.83 ±0.52 | 14.49 ±4.68 | 10.92 ±0.20 |
| **SparseFW** | 90.21 ±0.14 | 89.78 ±0.15 | 86.45 ±0.91 | 37.01 ±7.18 |
| **SSL** | 91.26 ±0.13 | 87.13 ±1.72 | 63.66 ±0.36 | 16.25 ±1.24 |
| **GLT** | 68.54 ±6.12 | 45.42 ±6.99 | 21.03 ±3.69 | 9.36 ±0.22 |
| **ABFP** | 91.45 ±0.49 | 91.43 ±0.52 | 91.17 ±0.42 | 28.91 ±0.71 |
| **SFP** | 88.90 ±3.78 | 67.66 ±17.25 | 28.44 ±12.21 | 10.55 ±1.33 |

Table 4: WideResNet on CIFAR-100: Comparison of Filter Pruning approaches. For each sparsity we indicate the achieved test accuracy after pruning averaged over all random seeds including standard deviation.

| | Sparsity | | | |
|---|---|---|---|---|
| **Method** | 10% | 20% | 30% | 40% |
| Baseline | 71.78 ±2.29 | 65.30 ±2.82 | 51.33 ±0.34 | 35.00 ±2.59 |
| **SparseFW** | 72.21 ±0.30 | 72.24 ±0.29 | 72.20 ±0.23 | 71.23 ±0.18 |
| **SSL** | 72.10 ±1.20 | 67.51 ±0.25 | 59.26 ±0.77 | 42.18 ±2.67 |
| **GLT** | 74.08 ±0.87 | 69.37 ±1.54 | 54.43 ±4.77 | 28.19 ±6.30 |
| **ABFP** | 71.49 ±0.12 | 71.50 ±0.17 | 71.53 ±0.11 | 71.51 ±0.13 |
| **SFP** | 73.45 ±0.35 | 73.47 ±0.34 | 73.05 ±0.04 | 63.82 ±2.14 |

Table 5: ResNet-50 on TinyImagenet: Comparison of Filter Pruning approaches. For each sparsity we indicate the achieved test accuracy after pruning averaged over all random seeds including standard deviation.

| | Sparsity | | | | | |
|---|---|---|---|---|---|---|
| Method | 10% | 20% | 30% | 40% | 50% | 60% |
| Baseline | 62.20 ±0.26 | 57.44 ±0.04 | 49.69 ±0.47 | 39.63 ±0.69 | 26.75 ±1.32 | 13.04 ±0.46 |
| **SparseFW** | 60.37 ±0.32 | 60.37 ±0.35 | 60.25 ±0.21 | 59.87 ±0.45 | 58.30 ±0.38 | 52.56 ±0.09 |
| **SSL** | 60.44 ±0.37 | 58.86 ±0.59 | 56.58 ±1.46 | 53.06 ±2.28 | 46.82 ±2.69 | 36.79 ±2.11 |
| **GLT** | 60.71 ±0.46 | 57.47 ±1.09 | 51.75 ±0.52 | 42.96 ±1.36 | 30.09 ±1.17 | 14.86 ±1.21 |
| **SFP** | 62.06 ±0.38 | 62.07 ±0.37 | 62.06 ±0.37 | 62.06 ±0.37 | 49.31 ±1.33 | 26.76 ±2.22 |
| **ABFP** | 60.28 ±0.63 | 60.31 ±0.69 | 60.28 ±0.58 | 60.20 ±0.59 | 60.20 ±0.49 | 59.99 ±0.55 |

Table 6: ResNet-50 on Imagenet: Comparison of Filter Pruning approaches. For each sparsity we indicate the achieved test accuracy after pruning averaged over all random seeds including standard deviation.

| | Sparsity | | | |
|---|---|---|---|---|
| **Method** | 10% | 20% | 30% | 40% |
| Baseline | 65.37 ±0.25 | 37.78 ±1.46 | 7.33 ±0.48 | 0.65 ±0.04 |
| **SparseFW** | 69.85 ±0.12 | 68.44 ±0.06 | 65.07 ±0.07 | 57.26 ±0.24 |
| **SSL** | 70.73 ±0.08 | 60.05 ±0.19 | 32.08 ±0.35 | 9.58 ±0.30 |
| **GLT** | 72.93 ±0.20 | 63.39 ±1.98 | 41.80 ±0.36 | 14.53 ±1.94 |
| **SFP** | 70.69 ±0.07 | 69.97 ±0.38 | 69.28 ±0.89 | 17.92 ±7.73 |
| **ABFP** | 70.54 ±0.23 | 69.33 ±0.33 | 66.29 ±1.56 | 59.27 ±0.39 |

### A.2.2 UNSTRUCTURED WEIGHT PRUNING

**CIFAR-10 Hyperparameter grids** For both the $k$-sparse polytope as well as $k$-support norm ball, we tune the fractional $k \in \{0.1, 0.2, 0.3, 0.4, 0.5\}$ and the multiplier $w \in \{10, 20, 30, 40, 50\}$ of the $L_2$-diameter.

Figure 4 compares the two feasible regions in an even larger hyperparameter search. The rows correspond to the $k$-sparse polytope (above) and $k$-support norm ball (below), respectively. The left column shows a heatmap of the test accuracy before pruning. While both approaches lead to well performing models for a wide range of hyperparameter configurations (indicated as the radius multiplier $w$ on the $x$-axis and $k$ on the $y$-axis), the $k$-support norm ball reaches higher results and converges properly for all configurations at stake. The $k$-sparse polytope approach fails to yield adequately trained dense models when the radius is relatively small but $k$ becomes larger, which is counter-intuitive, since larger $k$ allows a larger fraction of the parameters to be activated. The right column shows the corresponding heatmap of the test accuracy right after pruning. Clearly, the proposed approach is robust to pruning for a wider hyperparameter range.

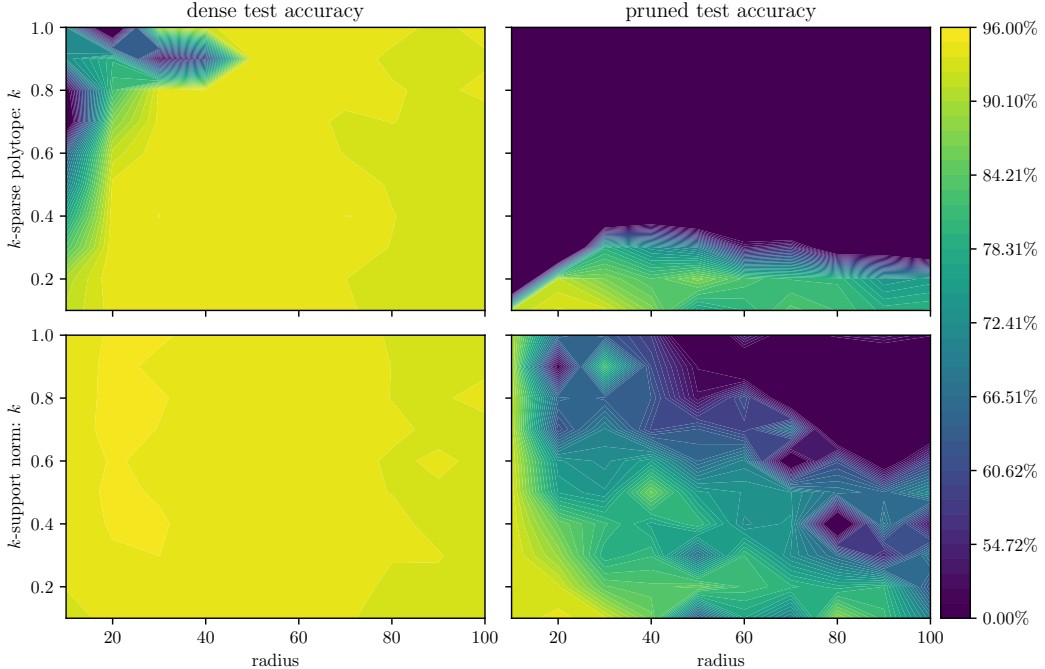

Figure 4: ResNet-18 on CIFAR-10: Contour plot when performing a large hyperparameter search over the radius and $k$ of the feasible regions, where the first row corresponds to the $k$-sparse polytope and the second one corresponds to the $k$-support norm ball. The left column shows the test accuracy before pruning, while the right column shows the test accuracy after pruning. The $k$-support norm approach leads to better performing dense models given the hyperparameter search at stake, which in turn are more stable to pruning.

Miao et al. (2022) showed that SFW (with $k$-sparse polytope constraints) outperforms SGD with weight decay, which in turn clearly, and unsurprisingly, outperforms the SFW-based approach when it is allowed to retrain. Our experiments indicate that while being less robust to pruning, SGD is able to reach on-par or better results after retraining, even when SFW is allowed to be retrained for the same amount of time. Leaving the domain of compression-aware training, this raises a more general question: in the case that retraining is not prohibited, is it beneficial to aim for robustness at pruning when trying to maximize the post-retraining accuracy?

Figure 5 illustrates an experiment where we investigate this exact question by performing One-Shot IMP (Han et al., 2015) to a sparsity of 95% and retraining for 10 epochs using LLR (Zimmer et al., 2021). We tuned both the weight decay for regular retraining as well as the weight decay for the

retraining phase. Non surprisingly, there is a weight decay sweet spot when it comes to maximizing the pre-pruning accuracy (left). The middle plot shows that higher weight decay typically leads to more robustness to pruning, however a too large weight decay hinders convergence of the dense model and might lower the performance after pruning. Surprisingly however, as depicted in the right plot showing the test accuracy after retraining, the optimal parameter configuration is the one that leads to the highest accuracy before pruning, which is also the least robust to pruning. This aligns with previous findings of Bartoldson et al. (2020), who question the strive for pruning stability when retraining is not prohibitive possible.

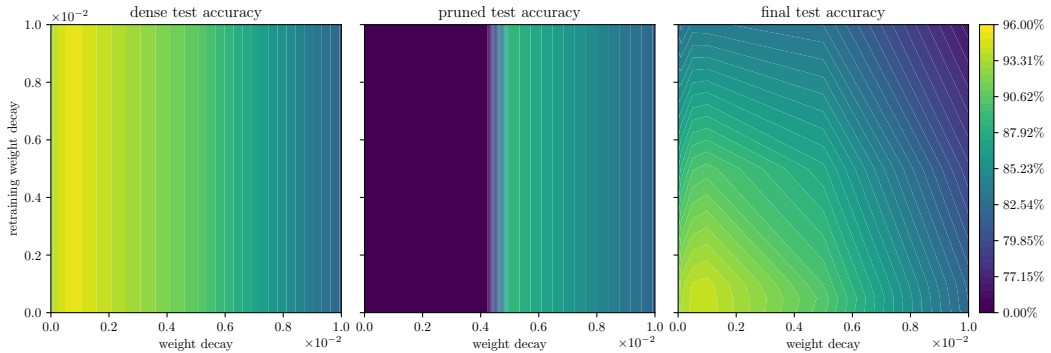

Figure 5: ResNet-18 on CIFAR-10: Test accuracy heatmap before pruning (left), after pruning (middle) and after retraining (right) when training SGD and applying One-Shot pruning, tuning both the weight decay during training ($x$-axis) as well as during retraining ($y$-axis).

### A.2.3 LOW-RANK MATRIX DECOMPOSITION

**CIFAR-10 Hyperparameter grids**  If not specified otherwise, we use weight decay values of {1e-4, 5e-4} for all algorithms except SparseFW.

- SparseFW: We tune the fractional $k \in \{0.1, 0.15, 0.2, 0.25, 0.3\}$ and the multiplier $w \in \{20, 30, 50\}$ of the $L_2$-diameter.
- NUC: We tune the nuclear norm penalty factor $\lambda \in \{1e-5, 5e-5, 1e-4, 5e-4, 1e-3, 8e-3, 5e-3, 1e-2, 5e-2\}$.
- SVDEnergy: We tune the nuclear norm thresholding $\lambda \in \{1e-2, 5e-2, 1e-1, 3e-1, 5e-1, 7e-1, 9e-1, 1e-0, 5e-0\}$.

**CIFAR-100 Hyperparameter grids**  If not specified otherwise, we use weight decay values of {1e-4, 5e-4} for all algorithms except SparseFW.

- SparseFW: We tune the fractional $k \in \{0.1, 0.15, 0.2, 0.25, 0.3\}$ and the multiplier $w \in \{20, 30, 50\}$ of the $L_2$-diameter.
- NUC: We tune the nuclear norm penalty factor $\lambda \in \{1e-6, 5e-6, 1e-5, 5e-5, 1e-4, 5e-4, 1e-3, 5e-3, 1e-2, 5e-2\}$.
- SVDEnergy: We tune the nuclear norm thresholding $\lambda \in \{1e-2, 5e-2, 1e-1, 3e-1, 5e-1, 7e-1, 9e-1, 1e-0, 5e-0\}$ and varied the weight decay in {1e-4, 2e-4, 5e-4}.

Table 7: ResNet-18 on CIFAR-10: Comparison of approaches for encouraging low-rank matrices throughout training. For each method we indicate the images-per-second throughput during training.

| | Sparsity | | | | | |
|---|---|---|---|---|---|---|
| **Method** | 40% | 50% | 60% | 70% | 80% | 90% |
| Baseline | 93.19 ±0.23 | 93.02 ±0.13 | 91.66 ±0.17 | 89.95 ±0.14 | 82.07 ±1.41 | 53.07 ±1.98 |
| **SparseFW** | 92.19 ±0.11 | 92.12 ±0.21 | 92.14 ±0.23 | 91.96 ±0.22 | 90.72 ±0.05 | 74.94 ±2.40 |
| **NUC** | 92.56 ±0.18 | 92.48 ±0.27 | 92.59 ±0.17 | 92.45 ±0.33 | 89.82 ±0.07 | 64.83 ±1.16 |
| **SVDEnergy** | 92.75 ±0.81 | 92.62 ±0.64 | 92.48 ±0.30 | 91.68 ±0.42 | 87.99 ±2.18 | 65.23 ±8.99 |

Table 8: WideResNet on CIFAR-100: Comparison of approaches for encouraging low-rank matrices throughout training. For each method we indicate the images-per-second throughput during training.

| | Sparsity | | | | | |
|---|---|---|---|---|---|---|
| **Method** | 30 % | 40% | 50% | 60% | 70% | 80% |
| Baseline | 75.57 ±0.10 | 74.28 ±2.59 | 73.52 ±0.95 | 64.67 ±11.12 | 56.04 ±2.24 | 10.97 ±8.29 |
| **SparseFW** | 75.53 ±0.18 | 75.69 ±0.04 | 75.75 ±0.08 | 75.37 ±0.40 | 75.30 ±0.10 | 73.28 ±2.02 |
| **NUC** | 75.96 ±0.29 | 74.86 ±1.53 | 72.35 ±0.48 | 68.57 ±4.95 | 55.97 ±2.80 | 5.40 ±2.12 |
| **SVDEnergy** | 75.69 ±0.95 | 75.14 ±0.18 | 74.11 ±0.81 | 66.76 ±4.41 | 55.40 ±3.04 | 30.82 ±26.87 |

### A.3 THE DYNAMICS OF GRADIENT RESCALING

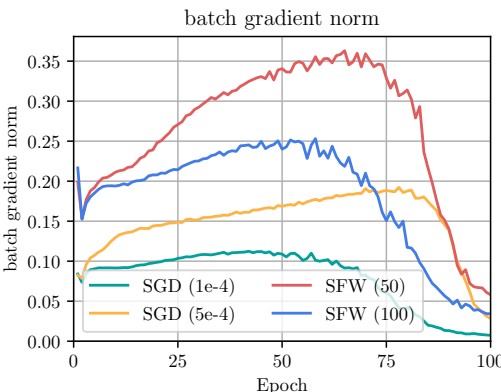

Figure 6: ResNet-18 on CIFAR-10: The evolution of the batch gradient norm $\|\nabla_t\|$ when training SFW for different values of $k$ and SGD for two weight decay strengths. The metric is averaged with respect to two random seeds and over all iterations within one epoch.

### A.4 PROOFS OF LMO CONSTRUCTIONS

In the following, we state the missing proof of the $k$-support norm LMO (being a special case of the group-$k$-support norm) and Lemma 3.1.

**Lemma A.1.** *Given $\nabla_t$, let $v_t \in \mathcal{C}_k(\tau) = conv\{v \mid \|v\|_0 \leq k, \|v\|_2 \leq \tau\}$ such that*

$$[v_t]_i = \begin{cases} -\tau[\nabla_t]_i/\|\nabla_t^{top_k}\|_2 & \textit{if } i \in top_k(|\nabla_t|), \\ 0 & \textit{otherwise}, \end{cases}$$

*where $\nabla_t^{top_k}$ is the vector obtained by setting to zero all $n-k$ entries $[\nabla_t]_j$ of $\nabla_t$ with $j \notin top_k(|\nabla_t|)$. Then $v_t \in \arg\min_{v\in\mathcal{C}_k(\tau)}\langle v, \nabla_t\rangle$ is a solution to Equation (2).*

*Proof.* By construction, all vertices $v$ of $\mathcal{C}_k(\tau)$ satisfy $\|v\|_2 = \tau$ and are $k$-Sparse, i.e., $\|v\|_0 \leq k$. Note that being $k$-Sparse includes cases where more than $n-k$ entries are zero. The minimum of

Equation (2) is attained at one such $v$. Further recall the following reformulation of the euclidean inner product:

$$\langle v, \nabla_t \rangle = \|v\|_2 \|\nabla_t\|_2 \cos(\angle(v, \nabla_t)) = \tau \|\nabla_t\|_2 \cos(\angle(v, \nabla_t)), \tag{10}$$

where $\angle(v, \nabla_t)$ denotes the angle between $v$ and $\nabla_t$. This term is minimized as soon as the angle between $v$ and $\nabla_t$ is maximal. If $v$ was not required to be $k$-Sparse, i.e., $v$ would be allowed to lie anywhere on the border of $B_2(\tau)$, the solution would clearly be given by $-\tau \nabla_t / \|\nabla_t\|_2$. However, since $v$ is $k$-Sparse, the vector maximizing the angle to $\nabla_t$ is the one that is closest to $-\tau \nabla_t / \|\nabla_t\|_2$ but is $k$-Sparse at the same time. This is exactly the one claimed. $\square$

**Lemma A.2.** *Given* $\nabla_t \in \mathbb{R}^{n \times m}$, *let* $\mathcal{W}_t \in \mathcal{C}_k^\sigma(\tau)$ *such that*

$$\mathcal{W}_t = \frac{-\tau}{\|\sigma(\Sigma_k)\|_2} U_k \Sigma_k V_k^T,$$

*where* $U_k \Sigma_k V_k^T$ *is the truncated SVD of* $\nabla_t$ *such that only the* $k$ *largest singular values are kept. Then* $\mathcal{W}_t \in \arg\min_{v \in \mathcal{C}_k^\sigma(\tau)} \langle v, \nabla_t \rangle$ *is a solution to Equation* (2).

*Proof.* Recall that

$$\mathcal{C}_k^\sigma(\tau) = \mathrm{conv}\{\mathcal{W} \in \mathbb{R}^{n \times m} \mid \mathrm{rank}(\mathcal{W}) \leq k, \|\sigma(\mathcal{W})\|_2 \leq \tau\}.$$

Let $\mathcal{W}$ be some minimizer. Note that rescaling a matrix by a scalar has no effect on its rank. Let us hence assume that $\|\sigma(\mathcal{W})\|_2 = \alpha$ for some $\alpha > 0$ and characterize $\mathcal{W} \in \arg\min_{\mathrm{rank}(v) \leq k} \langle v, \nabla_t \rangle$. Again, we have

$$\langle \mathcal{W}, \nabla_t \rangle_F = \langle \overleftarrow{\mathcal{W}}, \overleftarrow{\nabla_t} \rangle_2 = \alpha \|\overleftarrow{\nabla_t}\|_2 \cos(\angle(\overleftarrow{\mathcal{W}}, \overleftarrow{\nabla_t})), \tag{11}$$

where $\overleftarrow{x}$ is there vectorized form of matrix $x$. Since we can choose $\alpha \leq \tau$, this term is minimal as soon as the angle $\angle(\overleftarrow{\mathcal{W}}, \overleftarrow{\nabla_t})$ is maximal, i.e. $\cos(\angle(\overleftarrow{\mathcal{W}}, \overleftarrow{\nabla_t})) < 0$ and $\alpha = \tau$, where we use the same euclidean-geometric interpretation as in the proof for Lemma A.1 above. To obtain a maximal angle, we hence minimize the $L_2$-distance between $-\overleftarrow{\mathcal{W}}$ and $\overleftarrow{\nabla_t}$ in compliance with the rank constraint. Since again $\| - \overleftarrow{\mathcal{W}} - \overleftarrow{\nabla_t}\|_2 = \| - \mathcal{W} - \nabla_t\|_F$, the Eckart–Young–Mirsky theorem yields the claim, where we rescale appropriately to meet the Schattennorm constraint. $\square$

### A.5 CONVERGENCE OF SFW WITH GRADIENT RESCALING

Before priving the convergence of SFW with gradient rescaling as stated it informally in Theorem 5.1, we first recall some central definitions and assumptions.

#### A.5.1 SETTING

Let $\Omega$ be the set of training datapoints from which we sample uniformly at random. In Equation (1) we defined a unique loss function $\ell_i$ for each datapoint. In the following let $\ell(\theta, \omega_i) = \ell_i(\theta)$ for $\omega_i \in \Omega$. Similar to Reddi et al. (2016) and Pokutta et al. (2020), we define the SFW algorithm as follows, where the output $\theta_a$ is chosen uniformly at random from all iterates $\theta_0, \ldots, \theta_{T-1}$.

---

**Algorithm 1** Stochastic Frank–Wolfe (SFW)

---

**Input:** Initial parameters $\theta_0 \in \mathcal{C}$, learning rate $\eta_t \in [0, 1]$, batch size $b_t$, number of steps $T$.
**Output:** Iterate $\theta_a$ chosen uniformly at random from $\theta_0, \ldots, \theta_{T-1}$
1: **for** $t = 0$ to $T - 1$ **do**
2:     sample i.i.d. $\omega_1^{(t)}, \ldots, \omega_{b_t}^{(t)} \in \Omega$
3:     $\tilde{\nabla}L(\theta_t) \leftarrow \frac{1}{b_t} \sum_{j=1}^{b_t} \nabla\ell(\theta_t, \omega_j^{(t)})$
4:     $v_t \leftarrow \arg\min_{v \in \mathcal{C}} \langle \tilde{\nabla}L(\theta_t), v \rangle$
5:     $\theta_{t+1} \leftarrow \theta_t + \eta_t(v_t - \theta_t)$
6: **end for**

---

Let us recall some definitions. We denote the globally optimal solution by $\theta^\star$ and the *Frank–Wolfe Gap* at $\theta$ as

$$\mathcal{G}(\theta) = \max_{v \in \mathcal{C}} \langle v - \theta, -\nabla L(\theta) \rangle. \tag{12}$$

We will use the same assumptions as Reddi et al. (2016). First of all, let us assume that $L$ is $M$-smooth, that is

$$\|\nabla L(x) - \nabla L(y)\| \leq M\|x - y\| \tag{13}$$

for all $x, y \in \mathcal{C}$, which implies the well-known inequality

$$L(x) \leq L(y) + \langle \nabla L(y), x - y \rangle + \frac{M}{2}\|x - y\|^2. \tag{14}$$

Further, we assume the function $\ell$ to be $G$-Lipschitz, that is for all $x \in \mathcal{C}$ and $\omega \in \Omega$ we have

$$\|\nabla \ell(x, \omega)\| \leq G. \tag{15}$$

A direct consequence is that the norm of the gradient estimator can be bounded as $\|\tilde{\nabla} L(\theta_t)\| \leq G$.

### A.5.2 CONVERGENCE PROOF

The following well-established Lemma quantifies how closely $\tilde{\nabla} L(\theta)$ approximates $\nabla L(\theta)$. A proof can be found in Reddi et al. (2016).

**Lemma A.3.** *Let $\omega_1, \ldots, \omega_b$ be i.i.d. samples in $\Omega$, $\theta \in \mathcal{C}$ and $\tilde{\nabla} L(\theta) = \frac{1}{b} \sum_{j=1}^{b} \nabla \ell(\theta_t, \omega_j)$. If $\ell$ is $G$-Lipschitz, then*

$$\mathbb{E}\|\tilde{\nabla} L(\theta) - \nabla L(\theta)\| \leq \frac{G}{b^{1/2}}. \tag{16}$$

In the following, we denote the gradient estimator at iteration $t$ as $\nabla_t := \tilde{\nabla} L(\theta_t)$ and the $L_2$-diameter $\mathcal{D}(\mathcal{C})$ as $\mathcal{D}$. Let $\beta \in \mathbb{R}$ satisfy

$$\beta \geq \frac{2h(\theta_0)}{MD^2}, \tag{17}$$

for some given initialization $\theta_0 \in \mathcal{C}$ of the parameters, where $h(\theta_0) = L(\theta_0) - L(\theta^\star)$ denotes the optimality gap of $\theta_0$.

**Theorem A.4.** *For all $0 \leq t < T$, let $b_t = b = T$ and $\eta_t = \|\nabla_t\|\eta$ where $\eta = \left( \frac{h(\theta_0)}{TMD^2G^2\beta} \right)^{1/2}$. If $\theta_a$ is chosen uniformly at random from the SFW iterates $\{\theta_i : 0 \leq i < T\}$, then we have*

$$\mathbb{E}\left[ \mathcal{G}(\theta_a) \cdot \|\nabla L(\theta_a)\| \right] \leq \frac{D}{\sqrt{T}} \left( \sqrt{h(\theta_0)MG^2\beta} + G^2 + \frac{MGD}{2\sqrt{2}} \right),$$

*where $\mathbb{E}$ denotes the expectation w.r.t. all the randomness present.*

*Proof.* First of all notice that $\eta_t$ is well defined: Using $\beta$ as defined above we have

$$\eta \leq \left( \frac{1}{2TG^2} \right)^{1/2} = \frac{1}{G} \frac{1}{\sqrt{2T}} \tag{18}$$

and consequently we obtain $\eta_t = \|\nabla_t\|\eta \leq \frac{1}{\sqrt{2T}} \leq 1$ by using that $\|\nabla_t\| \leq G$. By $M$-smoothness of $L$ we have

$$L(\theta_{t+1}) \leq L(\theta_t) + \langle \nabla L(\theta_t), \theta_{t+1} - \theta_t \rangle + \frac{M}{2}\|\theta_{t+1} - \theta_t\|^2.$$

Using the fact that $\theta_{t+1} = \theta_t + \eta_t(v_t - \theta_t)$ and that $\|v_t - \theta_t\| \leq D$, it follows that

$$L(\theta_{t+1}) \leq L(\theta_t) + \eta_t \langle \nabla L(\theta_t), v_t - \theta_t \rangle + \frac{MD^2\eta_t^2}{2}. \tag{19}$$

Now let

$$\hat{v}_t = \arg\min_{v \in \mathcal{C}} \langle \nabla L(\theta_t), v \rangle = \arg\max_{v \in \mathcal{C}} \langle -\nabla L(\theta_t), v \rangle \tag{20}$$

be the LMO solution if we knew the exact gradient at iterate $\theta_t$, where $t = 0, \ldots, T-1$. This minimizer is not part of the algorithm but is crucial in the subsequent analysis. Note that we have

$$\mathcal{G}(\theta_t) = \max_{v \in \mathcal{C}} \langle v - \theta_t, -\nabla L(\theta_t) \rangle = \langle \hat{v}_t - \theta_t, -\nabla L(\theta_t) \rangle. \tag{21}$$

Continuing from Equation (19), we therefore have

$$L(\theta_{t+1}) \leq L(\theta_t) + \eta_t \langle \tilde{\nabla} L(\theta_t), v_t - \theta_t \rangle + \eta_t \langle \nabla L(\theta_t) - \tilde{\nabla} L(\theta_t), v_t - \theta_t \rangle + \frac{MD^2 \eta_t^2}{2}$$

$$\leq L(\theta_t) + \eta_t \langle \tilde{\nabla} L(\theta_t), \hat{v}_t - \theta_t \rangle + \eta_t \langle \nabla L(\theta_t) - \tilde{\nabla} L(\theta_t), v_t - \theta_t \rangle + \frac{MD^2 \eta_t^2}{2}$$

$$= L(\theta_t) + \eta_t \langle \nabla L(\theta_t), \hat{v}_t - \theta_t \rangle + \eta_t \langle \nabla L(\theta_t) - \tilde{\nabla} L(\theta_t), v_t - \hat{v}_t \rangle + \frac{MD^2 \eta_t^2}{2}$$

$$= L(\theta_t) - \eta_t \, \mathcal{G}(\theta_t) + \eta_t \langle \nabla L(\theta_t) - \tilde{\nabla} L(\theta_t), v_t - \hat{v}_t \rangle + \frac{MD^2 \eta_t^2}{2},$$

where the first inequality is just a reformulation of Equation (19) and the second one is due to the minimality of $v_t$. Applying Cauchy–Schwarz and using the fact that the diameter of $\mathcal{C}$ is $D$, we therefore have

$$L(\theta_{t+1}) \leq L(\theta_t) - \eta_t \, \mathcal{G}(\theta_t) + \eta_t D \| \nabla L(\theta_t) - \tilde{\nabla} L(\theta_t) \| + \frac{MD^2 \eta_t^2}{2}. \tag{22}$$

Now note that $\eta_t = \|\nabla_t\| \eta \leq G\eta$, yielding

$$L(\theta_{t+1}) \leq L(\theta_t) - \eta_t \, \mathcal{G}(\theta_t) + \eta G D \| \nabla L(\theta_t) - \tilde{\nabla} L(\theta_t) \| + \frac{MD^2 G^2 \eta^2}{2}. \tag{23}$$

Let $\theta_{0:t}$ denote the sequence $\theta_0, \ldots, \theta_t$. Taking expectations and applying Lemma A.3, we get

$$\mathbb{E}_{\theta_{0:t+1}} L(\theta_{t+1}) \leq \mathbb{E}_{\theta_{0:t+1}} L(\theta_t) - \mathbb{E}_{\theta_{0:t+1}} [\eta_t \mathcal{G}(\theta_t)] + \frac{DG^2 \eta}{b^{1/2}} + \frac{MD^2 G^2 \eta^2}{2}. \tag{24}$$

By rearranging and summing over $t = 0, \ldots, T-1$, we get the upper bound

$$\sum_{t=0}^{T-1} \mathbb{E}_{\theta_{0:t+1}} [\eta_t \mathcal{G}(\theta_t)] \leq L(\theta_0) - \mathbb{E}_{\theta_{0:T}} L(\theta_T) + \frac{TDG^2 \eta}{b^{1/2}} + \frac{TMD^2 G^2 \eta^2}{2}$$

$$\leq L(\theta_0) - L(\theta^\star) + \frac{TDG^2 \eta}{b^{1/2}} + \frac{TMD^2 G^2 \eta^2}{2}. \tag{25}$$

Now fix $t$ and apply the law of total expectation to reformulate

$$\mathbb{E}_{\theta_{0:t+1}} [\eta_t \mathcal{G}(\theta_t)] = \mathbb{E}_{\theta_{0:t}} \mathbb{E}_{\theta_{0:t+1}} [\eta_t \mathcal{G}(\theta_t) \mid \theta_{0:t}] = \mathbb{E}_{\theta_{0:t}} [\mathcal{G}(\theta_t) \eta \cdot \mathbb{E}_{\theta_{0:t+1}} [\|\nabla_t\| \mid \theta_{0:t}]], \tag{26}$$

where we exploited that once $\theta_{0:t}$ is available, $\mathcal{G}(\theta_t)$ is not subject to randomness anymore. The expected norm of the gradient estimator given $\theta_t$ depends only on the uniform selection of samples, allowing us to exploit the unbiasedness of the estimator as well as the convexity of the norm $\| \cdot \|$ using Jensen's inequality as follows:

$$\mathbb{E}_{\theta_{0:t+1}} [\|\nabla_t\| \mid \theta_{0:t}] = \mathbb{E}_\omega [\|\nabla_t\| \mid \theta_{0:t}] \tag{27}$$

$$\geq \| \mathbb{E}_\omega [\nabla_t \mid \theta_{0:t}] \| \tag{28}$$

$$= \| \frac{1}{b} \sum_{j=1}^{b} \mathbb{E}_{\omega_j} \nabla \ell(\theta_t, \omega_j) \| \tag{29}$$

$$= \| \nabla L(\theta_t) \|. \tag{30}$$

Combining this with Equation (25), we obtain

$$\eta \sum_{t=0}^{T-1} \mathbb{E}_{\theta_{0:t}} [\mathcal{G}(\theta_t) \cdot \|\nabla L(\theta_t)\|] \leq h(\theta_0) + \frac{TDG^2 \eta}{b^{1/2}} + \frac{TMD^2 G^2 \eta^2}{2}. \tag{31}$$

Using the definition of $\theta_a$, being a uniformly at random chosen iterate from $\theta_0, \ldots, \theta_{T-1}$, we conclude the proof with the following inequality.

$$\mathbb{E} [\mathcal{G}(\theta_a) \cdot \|\nabla L(\theta_a)\|] \leq \frac{h(\theta_0)}{T\eta} + \frac{DG^2}{b^{1/2}} + \frac{MD^2 G^2 \eta}{2} \tag{32}$$

$$\leq \frac{D}{\sqrt{T}} \left( \sqrt{h(\theta_0) MG^2 \beta} + G^2 + \frac{MGD}{2\sqrt{2}} \right) \tag{33}$$

$$\square$$

