# OpenReview forum: "Compression-aware Training of Neural Networks using Frank-Wolfe"
_ICLR.cc/2023/Conference — Submitted to ICLR 2023_

### Official Review · Reviewer_v7Db · 2022-10-22

**Confidence:** 4
**Clarity, Quality, Novelty And Reproducibility:** Quality and clarity are good. Origina…
**Correctness:** 4
**Technical Novelty And Significance:** 2
**Empirical Novelty And Significance:** 3
**Recommendation:** 6

**Strength And Weaknesses:**

Strength:
Overall, the paper is well-organized and easy to follow. This paper studies two optimization problems: convolutional filter pruning and low-rank matrix decomposition. For the first problem, the author uses the group-k-support norm ball to constrain the optimization problem to compress the model instead of the existing k-sparse approach. To solve this problem, the author proposes to use the SFW algorithm. For the second problem, the spectral-k-support norm is used. Experiments results on those two problems show the advantage of the proposed method. Specifically, the accuracy of the proposed method is more robust than the existing methods when compressing the model.

This work also empirically shows that the robustness of SFW can largely be attributed to the usage of the gradient rescaling of the learning rate. To justify the usage of gradient rescaling theoretically, the convergence of SFW with batch gradient dependent step size in the non-convex setting is established.

Weaknesses:
1.  In the abstract, the author said that compression-aware training could obtain state-of-the-art dense models which are robust to a wide range of compression ratios using a single dense training run while also avoiding retraining. But I believe the proposed framework still needs to run several times independently to obtain models with different compression ratios. That is, to obtain a model with different compression ratios, you have to run SFW another time from the beginning. Therefore, I am not sure why this method can avoid retraining.

2. I notice that In Figure 2, the performance of ABFP is comparable to or even better than SFW. Could the author show the advantage of SFW compared to ABFP?

3. Although Theorem 3.1 shows the convergence of SFW with batch gradient dependent step size in the non-convex setting.  However, the assumption of the objective function seems to be too strong (L-smooth and L-Lipschitz). I believe neither of the two studied problems would satisfy these assumptions. Please correct me if I am wrong.


**Summary Of The Paper:**

This work proposes the SFW algorithm to solve a constrained optimization framework. The proposed method can result in well-performing models that are robust towards convolutional filter pruning as well as low-rank matrix decomposition. Experiment results also show that the proposed method has better “accuracy vs sparsity” performance than existing approaches.

**Summary Of The Review:**

The experiment result is convincing. I also like the idea of combing the SFW and the group-k-support norm constraint together, but I didn’t find much novelty in the theory and algorithm respects.

---

> ### Author Response · Authors · 2022-11-18
> **Reply to Reviewer v7Db**
>
> We thank you for your detailed review and interest in our research. In the following, let us address your concerns.
> > In the abstract, the author said that compression-aware training could obtain state-of-the-art dense models which are robust to a wide range of compression ratios using a single dense training run while also avoiding retraining. But I believe the proposed framework still needs to run several times independently to obtain models with different compression ratios. That is, to obtain a model with different compression ratios, you have to run SFW another time from the beginning. Therefore, I am not sure why this method can avoid retraining.
>
> As we highlighted in our work, there is clearly a tradeoff between regularization strength and training performance. To give a simple example: When performing simple training with SGD and weight decay, a higher-than-usual weight decay will most likely harm the performance however increase the robustness towards unstructured magnitude pruning since the weights are tend to be closer to zero (cf. Figure 5 in the appendix, where we validate this through experiments). In short: When aiming for for robustness to high compression ratios, it is probably advisable to use a higher weight decay, albeit at the expense of losing a lot of performance when pruning smaller amounts from the model. To measure how different algorithms react to this tradeoff, we always plotted the hyperparameter configurations that lead to the highest on average performance among all sparsities. The point we are trying to make here (and that we show empirically) is that our proposed method performs well for a wide range of sparsities, that is, SFW is able to lead to high-performing dense models that are not only robust to a certain range of compression ratios, but for a decently large one. This separates our approach from others. In all experiments, no retraining is performed at all.
>
> Does this resolve your concerns or is there still an issue? We are happy to answer further questions if necessary.
>
> > I notice that In Figure 2, the performance of ABFP is comparable to or even better than SFW. Could the author show the advantage of SFW compared to ABFP?
>
> We agree and mentioned this in the discussion of the experimental results. ABFP is particularly strong in that regard. For CIFAR-10 and CIFAR-100, SparseFW and ABFP are comparable and mostly indistinguishable in their performance. For TinyImageNet and the recently added ImageNet we see slight advantages of ABFP over our approach, which we think does not stand in conflict with the claims we make.
>
> > Although Theorem 3.1 shows the convergence of SFW with batch gradient dependent step size in the non-convex setting. However, the assumption of the objective function seems to be too strong (L-smooth and L-Lipschitz). I believe neither of the two studied problems would satisfy these assumptions. Please correct me if I am wrong.
>
> These are in fact common assumptions, cf. e.g. Reddi et al.

---

### Official Review · Reviewer_XfiW · 2022-10-23

**Confidence:** 4
**Clarity, Quality, Novelty And Reproducibility:** This paper is well writen
**Correctness:** 3
**Technical Novelty And Significance:** 2
**Empirical Novelty And Significance:** 2
**Recommendation:** 3

**Strength And Weaknesses:**

The authors adopt a classic optimization algorithm (SFW) to perform structure compression-aware training. Before this work is accepted, several concerns should be addressed:
(i)	The first concern is about the novelty. This work heavily depends on the unstructed compression-aware training by extending the unstructured pruning to structured pruning setting. The novelty is discounted.
(ii)	The authors claims that the adopted SFW methods achieve significant speedup compated with nuclear-norm regularization based approach. In general, for nuclear norm regularized optimization, several SVD-free methods have been proposed [1]. We require the authors compare their proposed approach with more advanced optimization methods for solving nuclear norm regularized problem to show its effecgtiveness for a fair comparision. Furthermore, the real speedup ration should be reported to show the efficacy of the proposed algorithm.
(iii)	The current experiments restrict to CIFAR-10, CIFAR-100, Tiny-ImageNet. In general, ImageNet-1K is a standard benchmark to verify the effiectiveness of the compression methods. We recommend the author provide more experents on ImageNet-1k.
(iv)	About Theorem 5.1. Why does the term E[g*||Grad{L}||] indicate the convergence of the proposed approach? THe authors should gives more discussions on this convergence criteria.


[1] SVD-free Convex-Concave Approaches for Nuclear Norm Regularization, IJCAI, 2017.


**Summary Of The Paper:**

In this work, the authors adopt SWF (stochastic Frank-Wolfe) algorithm to perform compression-aware training. Prelimary experiments demonstrate the efficacy of the proposed approach.

**Summary Of The Review:**

See the comments above.

---

> ### Author Response · Authors · 2022-11-18
> **Reply to Reviewer XfiW**
>
> Thank you for your review. Please let us address your concerns in more detail.
> > (i) The first concern is about the novelty. This work heavily depends on the unstructed compression-aware training by extending the unstructured pruning to structured pruning setting. The novelty is discounted.
>
> We disagree that our work is not a novel contribution. Our work is the first to propose utilizing such constraints in the context of structured pruning and low-rank matrix decomposition of neural networks. In fact, the unstructured setting is a special case of the case we are utilizing here. Could you elaborate on how you think that this work depends on the unstructured compression-aware training? Pokutta et al. were the first authors to consider the sparsity-inducing capabilities of SFW, however their work was not related to pruning at all. In fact, they proposed the $k$-sparse polytope and showed that training a network with such constraints lead to a higher number of 'inactive' parameters. They however did not analyze how these networks behave when parameters are actually removed. In their ICLR2022 Spotlight paper, Miao et al. on the other hand used the $k$-sparse polytope in the context of unstructured pruning, using a variety of ideas introduced by Pokutta et al., that is, the constraint itself, the gradient rescaling of the learning rate, the self-adjusting learning rate approach, et cetera. Our research, which was concurrent to that of Miao et al., was to further develop these ideas in a proper way. To that end, we developed these approaches and showed the effectiveness of SFW in the context of structured compression and compression-aware training in general, significantly improving also the $k$-sparse approach by leveraging the $k$-support norm. Our work is also the first in that regard that theoretically justifies the usage of gradient rescaling of the learning rate, which we have shown to be essential to even obtain the sparsity-inducing behaviour of SFW.
>
> In that regard, we are unable to agree that our paper is insignificant. It is a novel contribution to the research community since we extend theoretical justifications of learning rate rescaling, and it is a technical contribution since we are the first to propose regularizing neural networks with the aforementioned constraints in the context of pruning and decomposition.
>
>
> > (ii) The authors claims that the adopted SFW methods achieve significant speedup compated with nuclear-norm regularization based approach. In general, for nuclear norm regularized optimization, several SVD-free methods have been proposed [1]. We require the authors compare their proposed approach with more advanced optimization methods for solving nuclear norm regularized problem to show its effecgtiveness for a fair comparision. Furthermore, the real speedup ration should be reported to show the efficacy of the proposed algorithm.
>
> We are happy to extend our work by more baselines in the compression-aware training setting. However, the paper you are referring to provides a method for an SVD-free Nuclear Norm Regularization in the case of **convex** loss functions. The loss of Neural Networks is highly non-convex. Could you elaborate more on how this might be a suitable baseline for our method? Regarding the speedups induced by our proposal, please see the official comment for a detailed account.
>
> > (iii) The current experiments restrict to CIFAR-10, CIFAR-100, Tiny-ImageNet. In general, ImageNet-1K is a standard benchmark to verify the effiectiveness of the compression methods. We recommend the author provide more experents on ImageNet-1k.
>
> We agree that ImageNet-1k is a necessary baseline in our setting. For that reason, we added Filter Pruning on ImageNet-1k to the current revision, please see the official comment.
>
> > (iv) About Theorem 5.1. Why does the term $E[g*|Grad{L}|$ indicate the convergence of the proposed approach? The authors should gives more discussions on this convergence criteria.
>
> In the case of unconstrained problems (e.g. those which are solvable using classical SGD), a typical measure for convergence is that the norm of the gradient vanishes over time, indicating convergence to a stationary point. For constrained optimization, this is impossible since a the feasible region might not contain such a point. To measure convergence of the FW-algorithm, one typically introduces the FW-Gap, which stems from the convex setting. In the non-convex setting, one has that this gap is zero if and only if a stationary point of the algorithm is reached. We show that in expectation the gap times the gradient norm vanishes, which is a rescaled version of the gap and clearly, this product vanishes if and only if the algorithm reaches a stationary point. We thank you for your remark and will update the paper to make this more clear.

---

### Official Review · Reviewer_gmRx · 2022-10-25

**Confidence:** 4
**Correctness:** 3
**Technical Novelty And Significance:** 2
**Empirical Novelty And Significance:** 2
**Recommendation:** 3

**Clarity, Quality, Novelty And Reproducibility:**

The writing can significantly be improved.
The novelty of this paper is very limited. This paper basically combines
the existing k-support norm regularztion (https://arxiv.org/pdf/1204.5043.pdf,
https://ieeexplore.ieee.org/stamp/stamp.jsp?arnumber=7952587) and the existing stochastic
Frank-Wolfe methods (https://arxiv.org/pdf/1607.08254.pdf).

**Strength And Weaknesses:**

Strength:

The proposed method applied on benchmark image-classification architectures and
datasets outperform the existing compression-aware approaches.  In the case of low-rank
matrix decomposition, the proposed method can require much less computational resources
than nuclear-norm regularization based approaches by requiring only a fraction of
the singular values in each iteration.

Weakness:

The novelty of this paper is very limited. This paper basically combines
the existing k-support norm regularztion (https://arxiv.org/pdf/1204.5043.pdf,
https://ieeexplore.ieee.org/stamp/stamp.jsp?arnumber=7952587) and the existing stochastic
Frank-Wolfe methods (https://arxiv.org/pdf/1607.08254.pdf).

**Summary Of The Paper:**

This paper proposed a constrained optimization framework based on a versatile family of norm constraints and the stochastic FrankWolfe (SFW) algorithm. The proposed method apply on benchmark image-classification architectures and datasets, and it yields competitive results, often outperforming existing compression-aware approaches.

**Summary Of The Review:**

The proposed method applied on benchmark image-classification architectures and
datasets outperform the existing compression-aware approaches.  In the case of low-rank
matrix decomposition, the proposed method can require much less computational resources
than nuclear-norm regularization based approaches by requiring only a fraction of
the singular values in each iteration.

Some Questions:

1. Why the proposed method can require much less computational resources
than nuclear-norm regularization based approaches by requiring only a fraction of
the singular values in each iteration ? It would be great if the authors would detail it.

2. In the experimental results, we hope to see the efficiency of the proposed methods.
Please give the results on test accuracy vs time.

3. There will be a strict upper limit of 9 pages for the main text of the submission,
with unlimited additional pages for citations (https://iclr.cc/Conferences/2023/CallForPapers).
This paper maybe not fit for this request.




--------------------------------------------------------------------------------------------------------------------
---------------------------------------------------------------------------------------------------------------------
The authors still did not solve my main concern- the limited novelty of this paper.
So I support to reject this paper.

---

> ### Author Response · Authors · 2022-11-18
> **Reply to Reviewer gmRx**
>
> We thank you for your review and will address your concerns directly.
>
> > The novelty of this paper is very limited.
>
> We disagree that our work is not a novel contribution. Our work is the first to propose utilizing such constraints in the context of structured pruning and low-rank matrix decomposition of neural networks. In fact, the $k$-support norm is a special case of the case we are utilizing here. Pokutta et al. were the first authors to consider the sparsity-inducing capabilities of SFW, however their work was not related to pruning at all. In fact, they proposed the $k$-sparse polytope and showed that training a network with such constraints lead to a higher number of 'inactive' parameters. They however did not analyze how these networks behave when parameters are actually removed. In their ICLR2022 Spotlight paper, Miao et al. on the other hand used the $k$-sparse polytope in the context of unstructured pruning, using a variety of ideas introduced by Pokutta et al., that is, the constraint itself, the gradient rescaling of the learning rate, the self-adjusting learning rate approach, et cetera. Our research, which was concurrent to that of Miao et al., was to further develop these ideas in a proper way. To that end, we developed these approaches and showed the effectiveness of SFW in the context of structured compression and compression-aware training in general, significantly improving also the $k$-sparse approach by leveraging the $k$-support norm. Our work is also the first in that regard that theoretically justifies the usage of gradient rescaling of the learning rate, which we have shown to be essential to even obtain the sparsity-inducing behaviour of SFW.
>
> In that regard, we are unable to agree that our paper is insignificant. It is a novel contribution to the research community since we extend theoretical justifications of learning rate rescaling, and it is a technical contribution since we are the first to propose regularizing neural networks with the aforementioned constraints in the context of pruning and decomposition.
>
> > The writing can significantly be improved.
>
> We take this concern very serious, however are unsure what parts can be improved. Is there any particular section that was not as easily accessible as intended?
>
> > Why the proposed method can require much less computational resources than nuclear-norm regularization based approaches by requiring only a fraction of the singular values in each iteration ? It would be great if the authors would detail it.
>
> Approaches based on nuclear-norm regularization require the full SVD to be computed. In each iteration, this may induce large computational overhead. Our approach relies only on the $k$ largest singular values, which can be computed efficiently if $k$ is not too large. We outlined this in section 3.2 and will make it more clear in the revision.
>
> > There will be a strict upper limit of 9 pages for the main text of the submission, with unlimited additional pages for citations. This paper maybe not fit for this request.
>
> Could you elaborate on why you think that we are violating this restriction? Our work is limited to exactly 9 pages, where we added a reproducibility statement right before the references. This is explicitly allowed, as stated in the call for papers: "The optional reproducibility statement will not count toward the page limit, but should not be more than 1 page."

---

### Official Review · Reviewer_RYk5 · 2022-10-27

**Confidence:** 3
**Correctness:** 4
**Technical Novelty And Significance:** 3
**Empirical Novelty And Significance:** 3
**Recommendation:** 8

**Clarity, Quality, Novelty And Reproducibility:**

The writing is clear and the ideas are easy to follow. The authors have agreed to release the code if accepted.

**Strength And Weaknesses:**

Pros:
- The proposed framework is interesting and beneficial to the community. The authors provide sufficient intuition and motivation behind the use of the sparsity-inducing norm constraints and how they can be effectively realized via SFW. The presentation is clear and the math is easy to follow.
- All the claims are supported well by empirical studies on benchmark datasets. The baselines seem sensible; although I must admit that I'm not too familiar with the related works in the compression-aware setting

Comments on the robustness study:
One of the interesting sections in the paper is the study on the robustness of the pruned model. The experimental study and the authors' discussion on the benefits of using the rescaled learning rate are insightful, especially at higher compression rates. With gradient scaling, the authors are able to show the 1/\sqrt(T) convergence result (FW gap) for SFW. However, they still collectively don't provide enough convincing arguments for the robustness claims, in my opinion.

**Summary Of The Paper:**

The paper proposes a novel framework for compression-aware training of neural networks. The proposed method uses norm constraints, for two types of pruning (1) convolutional filter pruning (2) low-rank matrix decomposition, expressed via updates of the Stochastic Frank-Wolfe (SFW) algorithm efficiently.

**Summary Of The Review:**

Overall, I think this is a good paper. The ideas presented are novel and are well-supported by ample experimental evidence and some theoretical results. The proposed method and discussions are relevant and useful to the community.

---

> ### Author Response · Authors · 2022-11-18
> **Reply to Reviewer RYk5**
>
> Thank you for your interest in our work. In the following, please let us address your concern regarding the strength of our experiments.
>
> > Comments on the robustness study: One of the interesting sections in the paper is the study on the robustness of the pruned model. The experimental study and the authors' discussion on the benefits of using the rescaled learning rate are insightful, especially at higher compression rates. With gradient scaling, the authors are able to show the 1/$\sqrt{T}$ convergence result (FW gap) for SFW. However, they still collectively don't provide enough convincing arguments for the robustness claims, in my opinion.
>
> The robustness towards sparsification lies at the heart of compression-aware training. We have shown empirically that SFW is perfectly suited for this setting, being robust towards compression - note that we do not perform retraining. We further added ResNet-50 trained on ImageNet to our latest revision in order to strengthen our claims, please see the official comment.
>
> Thanks again for taking the time to review our work, if you have further questions, we are happy to answer them.

---

### Author Response · Authors · 2022-11-18
**Official Revision Comment**

We like to thank all reviewers for evaluating our work and proposing potential directions for improvement. We appreciate considering our work to be "well written" (Reviewer XfiW), "easy to follow" (Reviewer v7Db) and above all "interesting and beneficial to the community" (Reviewer RYk5). To strengthen our claims, we added the following experimental setting to our revision:
- Filter Pruning ResNet-50 on ImageNet. Similarly to the previous results, SparseFW is able to be robust to filter pruning for a wide range of compression ratios, with only ABFP being on-par or better. We exchanged Plot 2c to represent the ImageNet results and added the full table to the appendix.

We again thank you for your suggestions and the interest in our work. In any case, we are happy to engage in further discussion in order to improve our work.

---

### Decision · Program_Chairs · 2023-01-20

**Decision:**

Reject

**Justification For Why Not Higher Score:**

The paper is generally of high-quality, but may require a careful revision (in the way the ideas are presented) to more clearly distinguish itself  from existing previous works.

**Justification For Why Not Lower Score:**

N/A

**Metareview: Summary, Strengths And Weaknesses:**

(a) Summary: The authors propose a new method for "compression-aware" training. This refers to a growing body of work which aims to train neural networks (NN) such that if the network weights were iteratively pruned post-hoc without any re-training, then performance would degrade slowly and gracefully. Conceptually this is achieved by updating the weights in a sparsity-aware manner. The contribution of this paper is to adapt the well-known (stochastic) Frank-Wolfe algorithm with (a) structured/group sparsity constraints, and (b) low-rank constraints for NN training. Experimental results show improvements over competing approaches that only use vanilla k-sparse constraints during training.

(b) Strengths: The paper is generally well-written. The idea of group-sparse constraints for training convnets is natural. The experiments are well constructed.

(c) Weaknesses: As pointed out by some reviewers, the central issue with the paper is novelty: the idea of using Frank-Wolfe for compression-aware training has been already proposed and validated thoroughly by Miao et al (ICLR '22). This significantly weakens the message of the current paper (starting from its title onwards).
To be clear, the authors do make a non-trivial contribution since they employ group-sparse/low-rank constraints, but Frank-Wolfe is such a general framework that merely changing the atomic set is somewhat marginal in my viewpoint. Algorithmically, this means that the linear minimization oracle needs to be adapted, but oracles for group-sparsity and low-rankness have long been established, so there too the conceptual innovation is somewhat marginal.

Overall recommendation: Reject. This is a good paper, which unfortunately, in the light of previous similar (published) approaches, does not meet the high bar set for acceptance.

**Summary Of Ac-Reviewer Meeting:**

Average scores placed it slightly below the "borderline bucket", so we didn't hold a meeting.